# Breaking the Factorization Barrier in Diffusion Language Models

**Ian Li** [* 1]   **Zilei Shao** [* 2]   **Benjie Wang** [2]
**Rose Yu** [1]   **Guy Van den Broeck** [2]   **Anji Liu** [3]

## Abstract

Diffusion language models theoretically allow for efficient parallel generation but are practically hindered by the "factorization barrier": the assumption that simultaneously predicted tokens are independent. This limitation forces a trade-off: models must either sacrifice speed by resolving dependencies sequentially or suffer from incoherence due to factorization. We argue that this barrier arises not from limited backbone expressivity, but from a structural misspecification: models are restricted to fully factorized outputs because explicitly parameterizing a joint distribution would require the Transformer to output a prohibitively large number of parameters. We propose **Co**upled **D**iscrete **D**iffusion (**CoDD**), a hybrid framework that breaks this barrier by replacing the fully-factorized output distribution with a lightweight, tractable probabilistic inference layer. This formulation yields a distribution family that is significantly more expressive than standard factorized priors, enabling the modeling of complex joint dependencies, yet remains compact enough to avoid the prohibitive parameter explosion associated with full joint modeling. Empirically, CoDD seamlessly enhances diverse diffusion language model architectures with negligible overhead, matching the reasoning performance of computationally intensive Reinforcement Learning baselines at a fraction of the training cost. Furthermore, it prevents performance collapse in few-step generation, enabling high-quality outputs at significantly reduced latencies. Code available at: https://github.com/liuanji/CoDD

---
[1]University of California, San Diego [2]University of California, Los Angeles [3]School of Computing, National University of Singapore. Correspondence to: Anji Liu <anjiliu@nus.edu.sg>.

*Proceedings of the $43^{rd}$ International Conference on Machine Learning*, Seoul, South Korea. PMLR 306, 2026. Copyright 2026 by the author(s).

## 1. Introduction

Diffusion language models (dLLMs) have recently emerged as a compelling paradigm for modeling natural language (Austin et al., 2021; Lou et al., 2024; Nie et al., 2025). Unlike traditional autoregressive language models that are bound to a fixed left-to-right generation order, dLLMs break this sequential constraint by offering the flexibility to make predictions in arbitrary orders and generate multiple tokens in parallel. By training a Transformer to predict distributions over sets of masked tokens simultaneously, dLLMs effectively enable global refinement of sequences, which offers a promising path toward bridging efficient parallel decoding with high-quality generation.

However, the parallel prediction capability of current dLLMs comes with a structural cost: it assumes that the simultaneously predicted tokens are mutually independent given the unmasked tokens (i.e., the context). When the model predicts a set of masked tokens in a single denoising step, it treats the probability of these tokens as the product of their univariate marginals (Liu et al., 2025a; Xu et al., 2025). This constraint arises from the nature of discrete state spaces. In continuous diffusion (Ho et al., 2020), dependencies are refined gradually via small updates to all variables. In contrast, dLLMs force the model to make hard commitments to multiple tokens in a single step. Since the model must choose these tokens simultaneously using a fixed context, it fails to account for how the choice of one predicted token should influence the others. Specifically, given a pre-defined parametric family $p_{\boldsymbol{\theta}}(\mathbf{X})$ for the remaining variables, the model approximates the conditional distribution by using a neural network $f$ to output a specific distribution parametrized by $\boldsymbol{\theta}$ based on context $\mathbf{c}$:

$$p(\boldsymbol{x}|\mathbf{c}) = p_{\boldsymbol{\theta}}(\boldsymbol{x})|_{\boldsymbol{\theta}=f(\mathbf{c})}.$$

The critical bottleneck in this formulation is the dimensionality of $\boldsymbol{\theta}$. Capturing even pairwise correlations would require parameters quadratic in the vocabulary size, which is computationally prohibitive. Consequently, $p_{\boldsymbol{\theta}}$ is typically restricted to fully factorized distributions, where $\boldsymbol{\theta}$ consists solely of independent logits for each position (Austin, 2018; Lou et al., 2024; Shi et al., 2024; Sahoo et al., 2024). This limitation forces a trade-off: highly dependent tokens must either be generated in separate steps or predicted simultane-

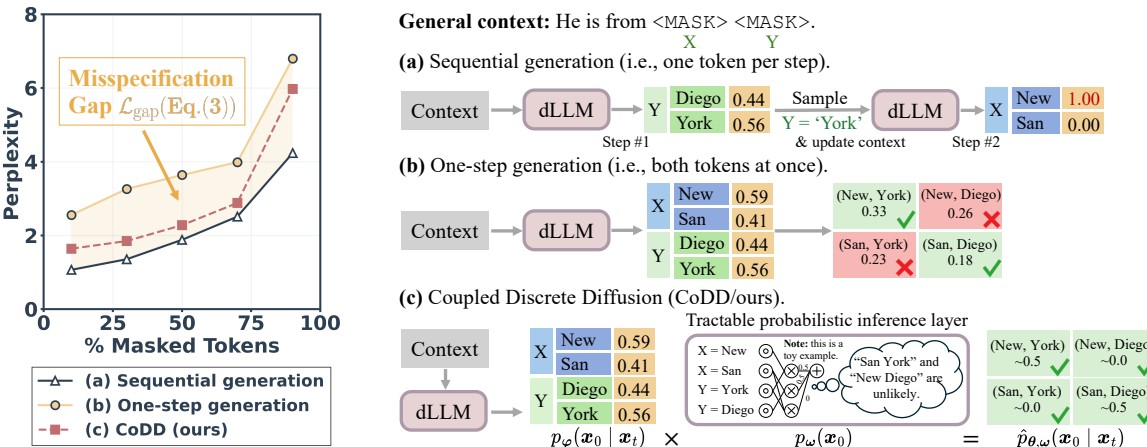

*Figure 1.* **Motivation and Intuition of CoDD. Left:** Illustration of the misspecification gap. The plot reports the perplexity of LLaDA (Nie et al., 2025) on the MathInstruct validation set across varying mask ratios. Curve **(a)** Sequential generation represents the ideal baseline (i.e., the true joint distribution learned by the model). When restricted to **(b)** One-step generation, the independence assumption causes significant performance degradation. The shaded region highlights this loss of perplexity, defined as the misspecification gap $\mathcal{L}_{\text{gap}}$. **(c)** CoDD significantly bridges this gap while retaining the efficiency of one-step prediction. **Right:** Conceptual comparison on "He is from `<MASK>` `<MASK>`". **(a)** Sequential generation accurately resolves dependencies but sacrifices speed. **(b)** One-step generation predicts in parallel but assumes independence, leading to incoherent mixtures like "San York". **(c)** CoDD overcomes this by modulating predictions with a tractable probabilistic inference layer, recovering valid joint distributions (e.g., "San Diego") in a single parallel step.

ously at the cost of incoherence.

In this paper, we argue that this dilemma is not an inherent limitation of parallel generation, but a symptom of structural misspecification. Even a hypothetical neural network $f$ with infinite capacity would fail to generate coherent parallel sequences if forced to bottleneck its knowledge through a fully factorized output distribution $p_{\theta}(\boldsymbol{x})$. To resolve this, we must **restructure the output itself to break the factorization barrier**: we require a distribution family that is *expressive* enough to capture complex joint dependencies, *compact* enough to be succinctly parameterized, and *inherently tractable* to support efficient probabilistic inference with arbitrary masking patterns.

We propose **Co**upled **D**iscrete **D**iffusion (**CoDD**), a hybrid framework that augments the Transformer backbone with a lightweight, *tractable probabilistic inference layer*. We instantiate this layer using Probabilistic Circuits (PCs) (Choi et al., 2020), a class of deep tractable models uniquely suited for this task. Crucially, PCs support the exact and efficient computation of marginal probabilities for any subset of variables, making them mathematically ideal for handling the arbitrary masking patterns inherent to diffusion. By "partially conditioning" the PC on the neural network's output $\theta$, we construct a parametric family that is significantly more expressive than existing fully factorized priors, yet maintains a compact parameter space for $\theta$. Further, this modular design ensures high training efficiency: the inference layer can be optimized either jointly with the neural backbone or separately as a lightweight, plug-and-play module.

Empirically, we demonstrate that CoDD seamlessly enhances diverse diffusion architectures (e.g., LLaDA, Dream) and decoding heuristics. Remarkable for its efficiency, CoDD can be trained in just ~3 GPU hours, less than 2% of the cost of competitive Reinforcement Learning (RL) baselines. This efficiency extends to inference, where the lightweight inference layer imposes minimal latency overhead. Despite this efficiency, CoDD matches or exceeds the reasoning performance of these expensive methods, boosting LLaDA's accuracy on MATH500 by +5.0% and Dream's performance on GSM8K by +10.8%. Furthermore, CoDD is robust in few-step generation, preventing performance collapse; for instance, it recovers GSM8K accuracy from 34.0% to 56.4% at 64 steps, enabling high-quality generation at a fraction of the standard cost.

## 2. Background

### 2.1. Diffusion Language Models

Diffusion language models (dLLMs) (Austin et al., 2021) generate samples through iterative token reconstruction. The process begins at step $T$ with a sequence $\boldsymbol{x}_T$ consisting entirely of `<MASK>` tokens, representing a state with no information. Over a sequence of time steps $t = T, \ldots, 1$, the model progressively "fills in the blanks", predicting subsets of the missing tokens in $\boldsymbol{x}_t$ to produce a refined sequence $\boldsymbol{x}_{t-1}$, until reaching $\boldsymbol{x}_0$, a complete and fully reconstructed state without any masking.

To perform iterative reconstruction, a neural network parameterized by $\varphi$ estimates token likelihoods by modeling the

conditional distribution of clean data given the current context $\boldsymbol{x}_t$, i.e., $p_{\boldsymbol{\varphi}}(\boldsymbol{x}_0|\boldsymbol{x}_t)$. To progress from step $t$ to $t-1$, we first sample a candidate reconstruction $\hat{\boldsymbol{x}}_0 \sim p_{\boldsymbol{\varphi}}(\cdot|\boldsymbol{x}_t)$. We then derive the next state $\boldsymbol{x}_{t-1}$ by re-masking this candidate via the posterior transition $q_{t-1}(\cdot|\hat{\boldsymbol{x}}_0, \boldsymbol{x}_t)$, which strictly preserves the visible tokens in $\boldsymbol{x}_t$ and re-masks the remaining positions with probability $\alpha_t \in (0, 1)$, thereby incrementally revealing the candidate prediction $\hat{\boldsymbol{x}}_0$.

To train the model, we simulate the partial contexts encountered during generation. We create training inputs $\boldsymbol{x}_t$ by taking a clean sample $\boldsymbol{x}_0 \sim p_{\text{data}}$ from the dataset and masking a subset of tokens following $\boldsymbol{x}_t \sim q_t(\cdot|\boldsymbol{x}_0)$, where $t \sim \text{Uniform}[1, \ldots, T]$ and each token is independently converted into <MASK> with probability $\alpha_t$. The model is then tasked to maximize the expected log-probability of the original sample $\boldsymbol{x}_0$ given the corrupted context:

$$\mathcal{L}(\boldsymbol{\varphi}) := \mathbb{E}_{t,\boldsymbol{x}_0,\boldsymbol{x}_t}\big[w(t) \cdot \log p_{\boldsymbol{\varphi}}(\boldsymbol{x}_0|\boldsymbol{x}_t)\big], \quad (1)$$

where $w(t)$ is a coefficient such that $\mathcal{L}(\boldsymbol{\varphi})$ forms an evidence lower-bound w.r.t. the masking distribution $q$ (Ou et al., 2024; Ye et al., 2025).

## 2.2. Sampling Algorithms

A main appeal of diffusion language models is their ability to predict tokens at arbitrary positions in parallel, bypassing the fixed left-to-right constraint of autoregressive decoding. That is, at every generation step, we can freely determine both the number of tokens to unmask and which specific ones to reveal.

However, while increasing the number of committed tokens in one step accelerates inference, doing so indiscriminately leads to substantial degradation in sample quality (Israel et al., 2025; Kim et al., 2025). To bridge this gap, recent decoding heuristics focus on identifying the most "reliable" positions to unmask. Strategies differ in how they define reliability: LLaDA commits the tokens with the highest likelihoods (Nie et al., 2025), while Dream prioritizes tokens with the lowest predictive entropy (Ye et al., 2025); more recently, margin-based decoding has been proposed to rank positions by the probability gap between the two most likely tokens (Kim et al., 2025).

## 3. The Cost of Parallel Prediction

Despite supporting arbitrary parallel generation in theory, discrete diffusion models exhibit a sharp trade-off between sample quality and efficiency. We argue that this limitation is not a matter of neural network backbone capacity, but a *structural misspecification* in their output parameterization: to maintain computational tractability, the denoising distribution $p_{\boldsymbol{\varphi}}(\boldsymbol{x}_0|\boldsymbol{x}_t)$ is constrained to be fully factorized, enforcing conditional independence among simultaneously

committed tokens (Liu et al., 2025a). As illustrated in Figure 1(b), this forces the joint probability of the reconstructed sequence to be modeled as a product of marginals:

$$p_{\boldsymbol{\varphi}}(\boldsymbol{x}_0 \mid \boldsymbol{x}_t) = \prod_{i=1}^{L} p_{\boldsymbol{\varphi}}(x_0^i \mid \boldsymbol{x}_t), \quad (2)$$

where $L$ is the sequence length. This factorization ignores the strong inter-token correlations inherent in language. By modeling the joint probability as a product of marginals, the model cannot capture multimodal dependencies. This inevitably leads to incoherent mixtures. For example, generating "San York" by confounding the distinct modes of "San Diego" and "New York" as demonstrated by Figure 1(b), yet the model has learned the correct dependencies, as evidenced by Figure 1(a) where the same backbone accurately recovers the valid modes (e.g., "New York") when restricted to sequential generation.

Following Liu et al. (2025a), we formalize this limitation as the *misspecification gap*, which is defined as the KL divergence between the ideal joint distribution (which captures the full dependencies encoded by the backbone) over $\mathbf{X}_0$ and the best possible factorized approximation (Eq. (2)):

$$\mathcal{L}_{\text{gap}} := \min_{\boldsymbol{\varphi}} D_{\text{KL}}(p_{\text{joint}}(\mathbf{X}_0|\boldsymbol{x}_t) \,\|\, p_{\boldsymbol{\varphi}}(\mathbf{X}_0|\boldsymbol{x}_t)). \quad (3)$$

We quantify this gap empirically in Figure 1(Left). The discrepancy between the sequential baseline ((a); representing $p_{\text{joint}}$) and the one-step parallel generation ((b); standing for $p_{\boldsymbol{\varphi}}$) reveals a substantial performance penalty attributable solely to the output constraint.

To keep this gap minimum, models are forced to unmask small and effectively independent subsets of tokens. Thus, valid inter-variable dependencies are captured only *sequentially*, effectively sacrificing the promised parallelism to circumvent the limited expressivity of the factorized output.

## 4. Unlocking Expressive Parallel Generation

We have established that the factorized output assumption imposes a structural ceiling on discrete diffusion models, forcing a trade-off between parallel efficiency and semantic coherence. To study this limitation, we distinguish between the model's contextual expressivity (the neural backbone's capacity) and its structural output constraints (the factorization required for tractability). Formalizing the perspective provided in the introduction, we decompose the denoising step $p_{\boldsymbol{\varphi}}(\boldsymbol{x}_0|\boldsymbol{x}_t)$ not as a monolithic operation, but as a composition of two distinct phases: parameter estimation and distribution modeling:

$$\boldsymbol{\theta} = f_{\boldsymbol{\varphi}}(\boldsymbol{x}_t), \quad \text{followed by} \quad \boldsymbol{x}_0 \sim p_{\boldsymbol{\theta}}(\mathbf{X}_0), \quad (4)$$

where $f_{\boldsymbol{\varphi}}$ is the neural network that maps the context $\boldsymbol{x}_t$ to a set of predictive parameters $\boldsymbol{\theta}$ (e.g., logits).

This decomposition reveals the precise locus of the mis-specification: while the parameter estimator $f_\varphi$ is highly expressive and encodes rich information of the context $\boldsymbol{x}_t$, the distribution $p_{\boldsymbol{\theta}}(\mathbf{X}_0)$ is structurally restricted to be fully factorized, as elaborated in the previous section.

A natural remedy is to replace the factorized product with a joint distribution. However, this encounters a severe bottleneck in the dimensionality of $\boldsymbol{\theta}$, which theoretically scales exponentially for arbitrary dependencies. Capturing even pairwise correlations would require parameters quadratic in the vocabulary size $V$, which is computationally prohibitive. The central challenge is thus to identify a distribution family that is *expressive yet compact*, capturing complex inter-token correlations while maintaining a parameter footprint comparable to the $L \times V$ logits.

### 4.1. Joint Modeling via Product Composition

To construct an output distribution that is both expressive and compact, we adopt a "base-and-refine" strategy. Rather than tasking the neural network $f_\varphi$ with constructing the entire dependency structure from scratch, we posit that the target conditional distribution can be decomposed into a structure-aware *global distribution* $p_{\boldsymbol{\omega}}(\boldsymbol{x}_0)$ and a context-aware *modulation* term $p_{\boldsymbol{\theta}}(\boldsymbol{x}_0)$, where the dependency on $\boldsymbol{x}_t$ is captured through $\boldsymbol{\theta}$.

We select multiplicative compositions over additive mixtures to leverage the representational power of intersecting densities. As established in the context of boosted generative models (Grover & Ermon, 2018), multiplicative interactions allow highly expressive distributions to be constructed from significantly simpler distributions. This allows the joint product to capture complex dependencies even when the modulation term $p_{\boldsymbol{\theta}}(\boldsymbol{x})$ remains structurally simple.

Formally, we model the denoising distribution $\hat{p}_{\boldsymbol{\theta},\boldsymbol{\omega}}(\boldsymbol{x}_0|\boldsymbol{x}_t)$ as the product of these two distinct components:

$$\hat{p}_{\boldsymbol{\theta},\boldsymbol{\omega}}(\boldsymbol{x}_0|\boldsymbol{x}_t) := \frac{1}{Z} \cdot p_{\boldsymbol{\omega}}(\boldsymbol{x}_0|\boldsymbol{x}_t) \cdot p_{\boldsymbol{\theta}}(\boldsymbol{x}_0), \qquad (5)$$

where $p_{\boldsymbol{\theta}}(\boldsymbol{x}_0)$ represents the fully factorized potentials predicted by the neural network (conditioned on $\boldsymbol{x}_t$), $p_{\boldsymbol{\omega}}(\boldsymbol{x}_0)$ is the learned structural prior, and $Z$ is the partition function.

However, this decomposition introduces a computational bottleneck: calculating the partition function $Z$ is generally intractable. We therefore require a prior $p_{\boldsymbol{\omega}}(\boldsymbol{x})$ that is expressive yet supports efficient normalization when multiplied by univariate potentials. This constraint directs us to Probabilistic Circuits (PCs) (Choi et al., 2020), a class of tractable models uniquely suited to satisfy this requirement, as we detail in the following section.

### 4.2. Tractable Inference with Probabilistic Circuits

Probabilistic Circuits (Choi et al., 2020) are a class of generative models that by design support efficient and exact computation of certain probabilistic queries (such as marginals) (Poon & Domingos, 2011; Vergari et al., 2021; Kisa et al., 2014). In this section, we define the syntax and semantics of PCs and demonstrate how they resolve the integration bottleneck identified in Equation (5).

**Definition 4.1** (Probabilistic Circuits). A PC $p(\boldsymbol{x})$ models a joint distribution over a set of variables $\mathbf{X}$ via a parameterized Directed Acyclic Graph (DAG) with a single root node $n_r$. The graph consists of input, product, and sum nodes. Input nodes define primitive distributions over some variable $X \in \mathbf{X}$, while sum and product nodes merge the respective input distributions of their children to build more complex distribution. Formally, the distribution encoded by every node $n$ is defined recursively as:

$$p_n(\boldsymbol{x}) := \begin{cases} g_n(\boldsymbol{x}) & n \text{ is an input node,} \\ \prod_{c \in \mathsf{ch}(n)} p_c(\boldsymbol{x}) & n \text{ is a product node,} \\ \sum_{c \in \mathsf{ch}(n)} \omega_{n,c} \cdot p_c(\boldsymbol{x}) & n \text{ is a sum node,} \end{cases} \quad (6)$$

where $\mathsf{ch}(n)$ denotes the children of node $n$, $g_n(\boldsymbol{x})$ is a univariate distribution (e.g., Categorical) over a variable $X \in \mathbf{X}$, and $\omega_{n,c}$ are learnable parameters associated with sum edges such that $\sum_{c \in \mathsf{ch}(n)} \omega_{n,c} = 1$. Intuitively, sum nodes and product nodes encode mixture and factorized distributions of their children, respectively. The set of trainable parameters $\boldsymbol{\omega}$ in a PC comprises the sum weights $\{\omega_{n,c}\}_{n,c}$ and the parameters defining the input distributions $\{f_n\}_n$.

The tractability of PCs hinges on applying the right structural constraints to their DAGs. Specifically, computing the partition function in Equation (5) necessitates a property known as *decomposability*. This constraint essentially requires that the children of any product node must model disjoint sets of variables. Intuitively, if we view the input nodes as representing symbolic variables, decomposability restricts the PC to encode multilinear polynomials, ensuring that no variable is ever multiplied by itself within a single product term. Please refer to Appendix C for details.

Answering probabilistic queries with PCs amounts to executing recursive algorithms in either bottom-up or top-down order over the graph structure. For example, computing the likelihood $p_{\boldsymbol{\omega}}(\boldsymbol{x})$ reduces to a single feedforward pass: we first evaluate the likelihood $p_n(\boldsymbol{x})$ at every input node given $\boldsymbol{x}$, and then propagate these probabilities upward through the sum and product nodes until reaching the root.

**Solving the Integration Bottleneck.** We now address the intractability of calculating the partition function $Z$ in Equation (5). While this theoretically requires summing over an exponential number of sequences, we can compute it efficiently by exploiting the decomposability of the PC.

Specifically, since the potentials coming from the neural network are fully factorized (i.e., $p_{\boldsymbol{\theta}}(\boldsymbol{x}_0) := \prod_i p_{\boldsymbol{\theta}}(x_0^i)$), we can "split" them to align with how the PC recursively divides the variables into disjoint sets. This pushes the global summation down to the leaves, breaking the exponential integral into tractable local computations.

Operationally, this allows us to compute $Z$ via a single feedforward pass. At each input node $n$ defined over variable $x_0^i$, we compute the local expectation of the neural potential under the input distribution, defined as $Z(n) := \sum_{x_0^i} g_n(x_0^i) \cdot p_{\boldsymbol{\theta}}(x_0^i)$. These values are then propagated upward, where the value of each sum and product node is computed following Equation (6). The global partition function $Z$ is then given by the value computed at the root node $n_r$, i.e., $Z = Z(n_r)$. We provide the formal justification for this algorithm in Appendix D.

### 4.3. Training Objective

We train CoDD using the standard discrete diffusion objective, modified to optimize our product distribution. Recall from Equation (1) that diffusion models maximize an ELBO with respect to the log-likelihood. We simply substitute the factorized distribution $p_{\boldsymbol{\varphi}}$ with our structurally augmented product distribution $\hat{p}_{\boldsymbol{\theta},\boldsymbol{\omega}}$:

$$\mathcal{L}(\boldsymbol{\omega}, \boldsymbol{\varphi}) := \mathbb{E}_{t,\boldsymbol{x}_0,\boldsymbol{x}_t}\left[w(t) \cdot \log \hat{p}_{\boldsymbol{\theta},\boldsymbol{\omega}}(\boldsymbol{x}_0|\boldsymbol{x}_t)\right]. \quad (7)$$

A key advantage of this decomposable parameterization is that it allows for modular training. Since the neural backbone $f_{\boldsymbol{\varphi}}$ that produces $\boldsymbol{\theta}$ (which parameterizes $p_{\boldsymbol{\theta}}(\boldsymbol{x}_0)$) is already trained to act as a context-aware potential function, we can freeze its parameters $\boldsymbol{\varphi}$ and optimize only the structural prior $\boldsymbol{\omega}$. This allows us to train the PC $p_{\boldsymbol{\omega}}(\boldsymbol{x}_0)$ efficiently by using the frozen Transformer as a fixed potential generator, thereby avoiding the high computational cost of backpropagating through the entire deep network.

## 5. Decoding with Coupled Discrete Diffusion

Recall from Section 2 that decoding in discrete diffusion models is an iterative process of selecting and unmasking tokens. To generate samples with CoDD, we maintain this standard workflow but introduce a key shift in the sampling step: instead of drawing candidates from the backbone's fully factorized potentials $p_{\boldsymbol{\theta}}$, we sample from the joint product distribution $\hat{p}_{\boldsymbol{\theta},\boldsymbol{\omega}}$ (Eq. (5)).[1] This modular substitution allows us to largely inherit the decoding strategies (e.g., masking schedules, prompt handling) of the baseline. In the following, we discuss the specific techniques adopted to effectively sample from this joint distribution.

---

[1]Similar to computing the partition function $Z$, we can sample from the joint distribution using one forward and one backward pass of the PC. Please refer to Appendix C for details.

### 5.1. Sampling Strategies

A critical component of effective generation in discrete diffusion models is *temperature scaling*. In standard formulations, given the fully factorized denoising distribution $p_{\boldsymbol{\theta}}$, we typically sample from its sharpened counterpart

$$p_{\boldsymbol{\theta},\tau}(\boldsymbol{x}_0) \propto p_{\boldsymbol{\theta}}^{1/\tau}(\boldsymbol{x}_0) = \prod_i p_{\boldsymbol{\theta}}^{1/\tau}(x_0^i),$$

where $\tau \in (0, 1]$ is the scaling factor. Unfortunately, extending this operation to our hybrid distribution $\hat{p}_{\boldsymbol{\theta},\boldsymbol{\omega}}$ is non-trivial, as renormalizing a PC after exponentiation is known to be #P-hard (Vergari et al., 2021). Consequently, we propose two tractable approximation methods.

**Latent Variable Sampling.** To circumvent the intractability of exact temperature scaling on the mixture distribution, we adopt an approximation based on the interpretation of PCs as deep latent variable models (Liu et al., 2023; Peharz et al., 2016). Formally, the circuit defines a joint distribution $\hat{p}_{\boldsymbol{\theta},\boldsymbol{\omega}}(\boldsymbol{x}) = \sum_{\boldsymbol{z}} \hat{p}_{\boldsymbol{\theta},\boldsymbol{\omega}}(\boldsymbol{x},\boldsymbol{z})$, where the discrete latent variables $\boldsymbol{z}$ represent the hierarchical routing decisions at sum nodes. The key insight is that conditioning on $\boldsymbol{z}$ resolves the intractable mixtures: once the latent path is fixed, the conditional distribution $\hat{p}_{\boldsymbol{\theta},\boldsymbol{\omega}}(\boldsymbol{x}|\boldsymbol{z})$ collapses into a single product term of leaf distributions. This allows us to apply temperature scaling exactly to the conditional component.

Specifically, we first sample a latent configuration $\boldsymbol{z} \sim \hat{p}_{\boldsymbol{\theta},\boldsymbol{\omega}}(\mathbf{Z})$, where the dependence on $\boldsymbol{x}_t$ is encapsulated in $\boldsymbol{\theta}$. We then sample the tokens $\boldsymbol{x}_0$ from the temperature-scaled conditional $\hat{p}_{\boldsymbol{\theta},\boldsymbol{\omega}}(\mathbf{X}_0|\hat{\boldsymbol{z}})^{1/\tau}$.

**Any-Order Autoregressive Sampling.** Alternatively, we can approximate the sharpened joint distribution by adopting a strategy akin to autoregressive decoding: sequentially determining tokens one by one. This allows us to apply standard temperature scaling to the conditional distribution of each individual token given the current context. Note that this sequentiality does not bind us to a fixed left-to-right order. We instead determine the unmasking order via the same reliability heuristics (e.g., highest confidence) utilized by the baseline. While this sequential refinement necessitates multiple queries to the structural prior, the computational overhead is minimal. Specifically, since the PC is significantly smaller than the Transformer backbone, this inner loop adds negligible latency, as we shall demonstrate in Table 3.

### 5.2. Diffusion Paradigms

Our framework is designed to be agnostic to the underlying diffusion strategy. We demonstrate this universality by integrating CoDD into two distinct paradigms: *Block Diffusion* (Arriola et al., 2025) and *Full Diffusion* (Austin et al., 2021). In the following, we detail the specific implementa-

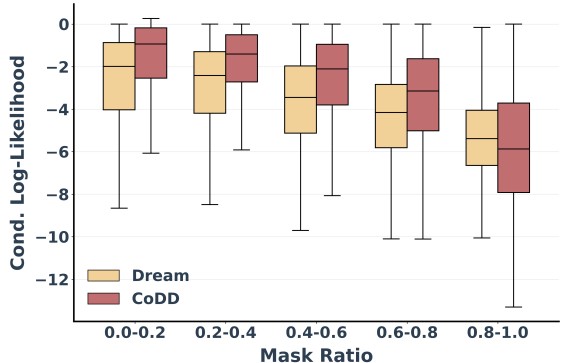

*Figure 2.* **Conditional Likelihood on Ground Truth.** This figure illustrates the conditional log-likelihood (CLL) of the CoDD and Dream models evaluated directly on ground truth question-answer pairs from the full MathInstruct dataset.

tion strategies adopted for each setting.

**Block Diffusion.** In this regime, the sequence is generated in fixed-size segments (e.g., $L_b = 32$), enforcing a semi-autoregressive order where all tokens in the preceding block are fully resolved before generation advances to the next. We therefore define the PC over sequences of length $L_b$. Within each active block, we generate candidates by sampling from the local joint distribution $\hat{p}_{\theta,\omega}$ using the temperature scaling techniques described in Section 5.1, effectively treating each block as a self-contained diffusion process conditioned on the visible history.

**Full Diffusion with Dynamic Windowing.** In this setting, the model denoises the entire sequence (e.g., 512 tokens) simultaneously. A practical challenge arises here: tractable PCs are typically trained on shorter contexts (e.g., 32) and cannot process the full global context at once. To resolve this, we introduce *Dynamic Windowing*. Specifically, we first utilize the baseline heuristic to select the subset of tokens for decoding. We then determine a fixed-size PC window to cover the maximum number of these tokens. This allows us to sample from the structurally guided joint distribution within an aligned local window, capturing the dependencies among the tokens that fall inside the window.

### 5.3. Adaptive Activation

In practice, we apply structural guidance adaptively: the PC is activated only when the mask ratio falls below a threshold $\gamma$. This design is grounded in our empirical analysis of the conditional log-likelihoods of CoDD.

As shown in Figure 2, CoDD exhibits a sharp performance crossover. In the low-noise regime (mask ratio < 0.8), CoDD assigns significantly higher probability to ground truth tokens, confirming that the PC effectively resolves local dependencies when context is available. Conversely, performance degrades in the high-noise regime (mask ratio

$\geq 0.8$), likely because the dependency structure of the data changes significantly throughout generation. Specifically, it shifts from abstract, global dependency to rigid, local correlations. A static PC, however, collapses these time-varying structures into a single global distribution. Consequently, it fails to offer precise guidance at any specific timestep, as the signal relevant to the current noise level is inevitably diluted by structural constraints from other regimes.

An interesting solution lies in enabling dynamic structural selection. Specifically, we can introduce control parameters output by the neural backbone to dynamically select or re-weight specific sub-distributions within the PC suitable for the current noise level. This would allow the structural prior to evolve in tandem with the diffusion process. We outline the mathematical formulation in Appendix E and identify this as a primary direction for future work.

## 6. Experiments on Reasoning Benchmarks

### 6.1. Models and Tasks

**Models.** We employ two instruction-tuned discrete diffusion language models: LLaDA-Instruct-8B (Nie et al., 2025) and Dream-Instruct-7B (Ye et al., 2025) as base models in our experiments. For LLaDA-Instruct-8B, we adopt block diffusion (LLaDA-block) at inference time, where the masked sequence is unmasked in fixed-size chunks, with diffusion processes applied locally within each active block. In contrast, Dream-Instruct-7B is evaluated under full diffusion, where progressive unmasking is applied over the full target sequence.

**Tasks.** We evaluate on four benchmarks spanning mathematical reasoning, scientific question answering, and code generation: MATH500 (Lightman et al., 2023), a subset consisting of 500 high-quality mathematical problems drawn from the Math dataset (Hendrycks et al., 2021), GSM8K (Cobbe et al., 2021), a dataset of grade school math problems for multi-step problem solving, GPQA (Rein et al., 2023) for challenging graduate-level question answering, and MBPP for program synthesis. All tasks are evaluated in a zero-shot setting. We provide more details about the experiment setup in Appendix B.

### 6.2. Training

We instantiate the structural prior $p_{\omega}(x)$ as a Probabilistic Circuit (Choi et al., 2020), specifically structured as a Hidden Markov Model (Rabiner & Juang, 1986) with a hidden state size of $N = 1024$.

The training procedure operates on a fixed dataset of question-solution pairs. For each sequence $x$, we sample a noise level $t \sim \mathcal{U}(0,1)$ and mask tokens in the solution segment with probability $t$, yielding a partially observed se-

*Table 1.* **Block Diffusion Performance Comparison (LLaDA).** Accuracy (%) of LLaDA baselines versus their CoDD-augmented versions. **Bold** indicates the best and underline indicates the second-best performance. Baselines use standard sampling strategies: [†]Margin (Kim et al., 2025), and [§]Low-Confidence masking (Chang et al., 2022).

| Model | Decoding Strategy / Diffusion Steps | MATH500 | | | GSM8K | | | GPQA | | | MBPP | | |
|---|---|---|---|---|---|---|---|---|---|---|---|---|---|
| | | 256 | 128 | 64 | 256 | 128 | 64 | 256 | 128 | 64 | 256 | 128 | 64 |
| LLaDA | Random | 28.40 | 19.20 | 7.40 | 58.83 | 41.32 | 14.86 | 22.10 | 19.20 | 10.71 | 18.00 | 6.20 | 2.20 |
| | Low Confidence[§] | 36.00 | 31.60 | 12.60 | 71.34 | 64.22 | 32.37 | 21.43 | 17.86 | 8.04 | 34.80 | 17.00 | 5.40 |
| | Margin[†] | 39.00 | 34.40 | 14.60 | 71.65 | 66.41 | **40.41** | 22.10 | 17.19 | 8.93 | 35.20 | 21.60 | 7.80 |
| CoDD | Random | 30.40 | 20.80 | 8.80 | 58.23 | 42.08 | 15.31 | **26.34** | **21.65** | **11.38** | 19.40 | 9.40 | 4.00 |
| | Δ *performance* | +2.00 | +1.60 | +1.40 | -0.61 | +0.76 | +0.45 | +4.24 | +2.46 | +0.67 | +1.40 | +3.20 | +1.80 |
| | Low Confidence | **41.00** | 33.80 | 15.80 | **72.63** | 65.88 | 34.65 | 24.55 | 17.86 | 9.15 | 34.00 | **23.80** | 7.20 |
| | Δ *performance* | +5.00 | +2.20 | +3.20 | +1.29 | +1.67 | +2.28 | +3.13 | 0.00 | +1.12 | -0.80 | +6.80 | +1.80 |
| | Margin | 39.20 | **38.80** | **18.40** | 72.48 | **66.72** | **40.41** | 25.89 | 18.53 | 10.94 | **35.60** | 22.80 | **8.80** |
| | Δ *performance* | +0.20 | +4.40 | +3.80 | +0.83 | +0.30 | 0.00 | +3.79 | +1.34 | +2.01 | +0.40 | +1.20 | +1.00 |

*Table 2.* **Full Diffusion Performance Comparison (Dream).** Accuracy (%) of Dream baselines versus their CoDD-augmented versions. **Bold** indicates the best and underline indicates the second-best performance. Baselines use standard sampling strategies: [†]Margin (Kim et al., 2025), [‡]Low-confidence (Chang et al., 2022), and [◇]Entropy (Ye et al., 2025).

| Model | Decoding Strategy / Diffusion Steps | Math 500 | | | GSM8K | | | GPQA | | | MBPP | | |
|---|---|---|---|---|---|---|---|---|---|---|---|---|---|
| | | 256 | 128 | 64 | 256 | 128 | 64 | 256 | 128 | 64 | 256 | 128 | 64 |
| Dream | Random | 16.20 | 17.00 | 16.20 | 38.36 | 39.50 | 36.01 | **27.90** | 24.78 | 25.00 | 29.80 | 32.20 | **29.40** |
| | Margin[†] | 32.20 | 21.80 | 10.80 | 70.43 | 56.10 | 35.70 | 24.33 | 14.06 | 8.04 | 45.60 | 32.20 | 23.60 |
| | Low Confidence[‡] | 21.80 | 18.40 | 12.60 | 38.36 | 46.85 | 40.11 | 27.01 | **27.68** | **26.12** | 42.00 | 35.60 | 26.60 |
| | Entropy[◇] | 32.20 | 21.80 | 10.80 | 71.34 | 56.18 | 33.97 | 24.33 | 14.06 | 8.04 | 47.20 | 36.20 | 24.20 |
| CoDD | Entropy | **36.80** | **25.80** | **18.20** | **74.75** | **67.02** | **56.41** | **27.90** | 18.08 | 10.94 | **47.40** | **36.60** | 27.80 |
| | Δ *performance* | +4.60 | +4.00 | +7.40 | +3.41 | +10.84 | +22.44 | +3.57 | +4.02 | +2.90 | +0.20 | +0.40 | +3.60 |

quence $x_t$. To isolate the structural learning task, we freeze the base diffusion model $f_\varphi$ and precompute its logits (i.e., $\theta$) for all training examples, which is used to compute the likelihood $\hat{p}_{\theta,\omega}(x_0|x_t)$. The PC parameters $\omega$ are then optimized using an **em**one (Liu et al., 2025b), a specialized optimizer for Probabilistic Circuits, to maximize the evidence-weighted conditional log-likelihood (i.e., Eq. (7)).

### 6.3. Results

**CoDD consistently improves performance across architectures and heuristics.** Table 1 and 2 report the performance of base models versus their CoDD-augmented counterparts. CoDD yields robust gains across all settings, acting as a universal booster regardless of the underlying diffusion paradigm (Block vs. Full). On LLaDA, CoDD amplifies the strong "Low Confidence" baseline, improving accuracy on Math 500 by **+5.00%** (256 steps) and MBPP by **+6.80%** (128 steps). On Dream, which utilizes full-sequence diffusion, the gains are even more pronounced: applying CoDD on top of the "Entropy" strategy boosts GSM8K accuracy from 56.18% to **67.02%** (+10.84%) at 128 steps. We provide additional results on Dream, as well as ablation studies on the adaptive activation threshold $\gamma$ and the temperature $\tau$ in Appendix B. See Appendix F for qualitative results.

**Recovering capabilities in low-compute regimes.** Stan-

dard diffusion models suffer catastrophic performance degradation when the number of denoising steps is reduced. CoDD mitigates this collapse, effectively maintaining reasoning capabilities in efficiency-constrained settings. In combination with Any-Order (AO) autoregressive sampling (Sec. 5.1), CoDD further extends these gains, elevating accuracy to 17.0% at 64 steps compared to LLaDA's 12.6%.

**Inference Latency Analysis.** To quantify inference-time efficiency, we evaluate the wall-clock time per sample across varying diffusion steps (64, 128, 256). As detailed in Table 3, CoDD introduces only a marginal latency overhead compared to the base architectures. Specifically, CoDD only spends **4-5%** across all budgets on the Dream backbone. This confirms that CoDD achieves the performance gains with minimal latency overhead, effectively preserving the rapid inference speeds inherent to diffusion language models. While AO introduces a small overhead increase compared to standard CoDD, it remains significantly more efficient than RL baselines.

**Training Efficiency.** Unlike standard finetuning methods, training the PC structural prior is computationally lightweight. The PC is trained on the frozen activations of the base model, requiring orders of magnitude less compute than methods involving backpropagation through the Transformer backbone. As shown in Figure 3, training CoDD

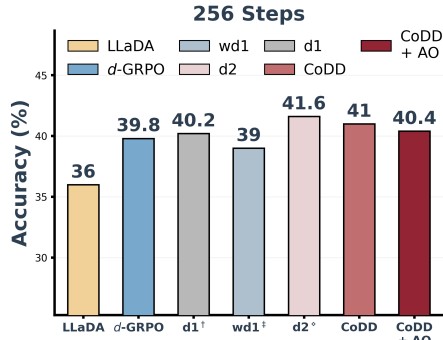 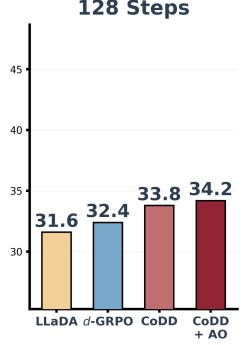 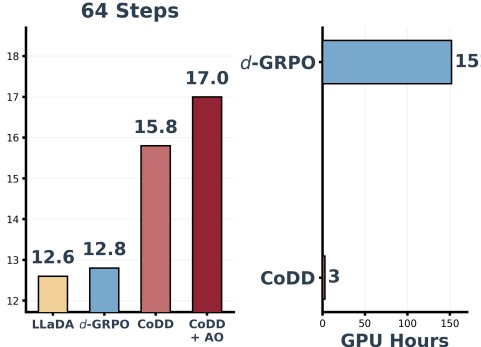

*Figure 3.* **Left: Performance Comparison on MATH500 vs. RL Baselines for 256/128/64 diffusion steps with a fixed generation length of 512.** *d*-GRPO denotes *diffu*-GRPO and is reproduced with Zhao et al. (2025)'s codebase. Methods marked with superscripts (d1[†], wd1[‡], d2[◇]) are reported from prior work (Zhao et al., 2025; Tang et al., 2025; Wang et al., 2025). **Right: Training-time cost in GPU hours (↓ is better) for *diffu*-GRPO and CoDD under our implementation.**[2]

*Table 3.* **Inference cost (seconds/sample) and overhead.** We report inference cost for each method at diffusion steps 64/128/256 on MATH500 and GPQA. Overhead is computed relative to the corresponding baseline (LLaDA or Dream) at the same step.

| Method | MATH500 | | | GPQA | | |
|---|---|---|---|---|---|---|
| | 64 | 128 | 256 | 64 | 128 | 256 |
| **LLaDA** | 2.86 | 5.49 | 11.51 | 3.11 | 6.21 | 12.44 |
| **CoDD** | 3.04 | 6.18 | 12.11 | 3.19 | 6.61 | 13.03 |
| *Overhead* | **6.29%** | **12.57%** | **5.21%** | **2.57%** | **6.44%** | **4.74%** |
| **CoDD + AO** | 3.63 | 6.67 | 12.16 | – | – | – |
| *Overhead* | 26.92% | 21.49% | 5.65% | – | – | – |
| *diffu*-**GRPO** | 3.81 | 7.64 | 15.31 | – | – | – |
| *Overhead* | 33.22% | 39.16% | 33.01% | – | – | – |
| **Dream** | 2.89 | 5.86 | 11.73 | 3.14 | 6.36 | 12.76 |
| **CoDD** | 3.00 | 6.11 | 12.35 | 3.29 | 6.68 | 13.33 |
| *Overhead* | **3.81%** | **4.27%** | **5.29%** | **4.78%** | **5.03%** | **4.47%** |

*Table 4.* **Open-Ended Text Generation on WikiText.** CoDD applied on top of Dream (low-confidence decoding). Lower PPL and NLL/tok are better.

| Steps | Method | PPL↓ | Div | Rep-2 | Rep-4 | NLL/tok↓ |
|---|---|---|---|---|---|---|
| 16 | **Dream** | 3.32 | 0.217 | 0.783 | 0.707 | 1.2013 |
| | *+ CoDD* | **2.92** | 0.188 | 0.812 | 0.747 | **1.0707** |
| 32 | **Dream** | 4.67 | 0.341 | 0.659 | 0.571 | 1.5403 |
| | *+ CoDD* | **4.08** | 0.316 | 0.684 | 0.599 | **1.4060** |
| 64 | **Dream** | 4.66 | 0.484 | 0.516 | 0.401 | 1.5397 |
| | *+ CoDD* | **4.15** | 0.463 | 0.537 | 0.427 | **1.4228** |

requires only ∼3 GPU hours to converge, a negligible fraction ($< 2\%$) of the compute budget typically demanded by RL-based methods like (Zhao et al., 2025). This makes CoDD a highly practical "plug-and-play" module for enhancing existing pre-trained diffusion models.

# 7. Experiments on Natural Language

## 7.1. Open-Ended Text Generation

We assess the quality of open-ended text generation produced by CoDD on the WikiText-103 validation set. For each evaluation instance, we randomly truncate a sequence to serve as a conditioning prefix, and the model is tasked with generating its continuation. The base model is Dream with low-confidence decoding, and the PC is trained on a corpus of samples drawn from a pre-trained GPT-2 model. We measure generation quality along several axes: perplexity (PPL) under a reference language model, lexical diversity (Div), $n$-gram repetition rates (Rep-2, Rep-4), and average negative log-likelihood per token (NLL/tok).

As reported in Table 4, CoDD consistently improves over

the Dream baseline across all step budgets. Perplexity decreases by 0.40, 0.59, and 0.51 points at 16, 32, and 64 steps respectively, while NLL/tok drops by a comparable margin. These gains demonstrate that the joint distribution captured by the PC produces continuations that are more probable under a strong autoregressive reference, indicating better-formed text. Repetition metrics (Rep-2, Rep-4) increase slightly under CoDD; this is consistent with the model converging onto higher-density regions of the data distribution, where natural recurrence of common $n$-grams is expected. The corresponding diversity (Div) values remain close to the baseline, suggesting that CoDD does not collapse to degenerate outputs but rather refines the model's predictions toward more coherent sequences.

**Likelihood Evaluation.** To directly quantify how well the joint distribution induced by CoDD matches natural language, we evaluate the conditional log-likelihood (LL) assigned to ground-truth completions on the WikiText-103 validation set. Following Discrete Copula Diffusion (DCD) (Liu et al., 2025a), we adopt SEDD (Lou et al., 2024) as the base diffusion model and train the PC on samples generated by GPT-2. For each instance, the model is conditioned on a randomly truncated prefix and assigns a probability to the true continuation. We bucket the evaluation by

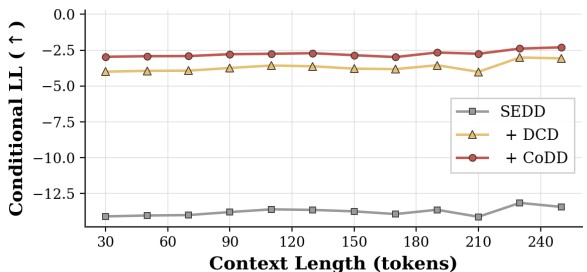

*Figure 4.* **Conditional Log-Likelihood on WikiText-103.** Conditional log-likelihood ($\uparrow$) of ground-truth completions as a function of context length, with SEDD as the base diffusion model.

context length (in tokens) and report mean log-likelihood per bucket.

Figure 4 compares the base SEDD model, DCD, and CoDD across context lengths ranging from 20 to 256 tokens. Two trends are evident. First, both DCD and CoDD dramatically improve over the fully-factorized SEDD baseline, with log-likelihoods rising from approximately $-14$ to $-4$ to $-3$. This confirms that explicitly modeling inter-token dependencies recovers a substantial amount of probability mass that the factorized parameterization is structurally incapable of expressing. Second, CoDD consistently outperforms DCD across every context-length bucket, with the average improvement on the order of 1 nat per token. Notably, CoDD achieves this with a far lighter computational footprint than DCD's autoregressive auxiliary model, reinforcing the practicality of Probabilistic Circuits as a tractable, lightweight inference layer for dLLMs.

## 8. Related Work

**Diffusion Language Models.** The adaptation of diffusion models to the discrete text domain began with D3PM (Austin et al., 2021), which formulated the forward process using stochastic transition matrices over categorical states. This foundation was extended by SEDD (Lou et al., 2024), which introduced a continuous-time framework to better model the discrete data distribution. A practical advancement came with Masked Diffusion Language Models (MDLM) (Sahoo et al., 2024) and MD4 (Shi et al., 2024), which derive a simplified training objective for masked diffusion models. Building on this masking paradigm, recent approaches like LLaDA (Nie et al., 2025) and Dream (Ye et al., 2025) have successfully scaled discrete diffusion to billions of parameters, demonstrating that non-autoregressive models can achieve competitive quality with autoregressive Transformers.

**The Factorization Barrier.** The limitations imposed by the independence assumption have been observed by several prior works (Kim et al., 2025; Liu et al., 2025a; Hayakawa et al., 2025). Initial efforts to mitigate this fo-

cused on confidence-based decoding, a heuristic strategy that prioritizes committing the most confident tokens (Chang et al., 2022; Ye et al., 2025). While this effectively mitigates the problem when the number of simultaneously predicted tokens is small, it becomes significantly less effective as we enter the few-step generation regime where the model must make aggressive parallel commitments. Alternative approaches have attempted to explicitly capture these dependencies by augmenting diffusion with other deep generative models, including autoregressive (Liu et al., 2025a) and energy-based models (Xu et al., 2025). However, the use of such additional models incurs significant computational overhead. In this work, we take a completely different route: we identify the root cause as a misspecification of the distribution class itself and employ Probabilistic Circuits as a lightweight, rigorous solution to mitigate this barrier.

**Applications of Probabilistic Circuits and HMMs.** Prior research has demonstrated the utility of tractable probabilistic models in language tasks, including efforts to scale Hidden Markov Models for language modeling (Chiu & Rush, 2020) and utilizing tensor decompositions to improve multi-token prediction (Basharin et al., 2024). In controlled generation, PCs provide tractable guidance for both autoregressive language models (Zhang et al., 2024; 2023) and continuous diffusion models for tasks like image inpainting (Liu et al., 2024).

Most closely related to our hybrid parameterization are architectures like Conditional Sum-Product Networks (Shao et al., 2020) and Probabilistic Neural Circuits (Dos Martires, 2024), which leverage deep neural networks to directly output or condition the internal parameters of the tractable circuit. Instead of conditionally predicting all circuit parameters, which scales poorly to large vocabularies, CoDD efficiently captures dependencies by tractably multiplying a structural PC prior with a fully factorized neural output.

## 9. Conclusion

In this work, we attributed the *factorization barrier* as a fundamental bottleneck in discrete diffusion models, preventing them from fully exploiting the efficiency of parallel generation without sacrificing coherence. We proposed Coupled Discrete Diffusion (CoDD), a hybrid architecture that breaks this barrier by augmenting the Transformer with a tractable Probabilistic Circuit to capture complex dependencies. This approach reconciles parallel efficiency with semantic coherence, enabling high-quality generation even in few-step regimes.

---

[2]GPU-hour comparison was measured on NVIDIA RTX PRO 6000 Blackwell Server Edition GPUs.

## Impact Statement

This paper presents work whose goal is to advance the field of Machine Learning. There are many potential societal consequences of our work, none which we feel must be specifically highlighted here.

## Acknowledgements

This work was funded in part by the National University of Singapore under its Start-up Grant (Award No: SUG-251RES250); this work was also funded in part by the DARPA ANSR and CODORD programs under awards FA8750-23-2-0004 and HR00112590089, and gifts from Cisco Research, Qualcomm, and Amazon. Additionally, this work is supported in part by the U.S. Army Research Office under Army-ECASE award W911NF-07-R-0003-03, the U.S. Department Of Energy, Office of Science, ARPA-H-SOL-24-101 program, IARPA HAYSTAC Program, DARPA YFA, NSF Grants #2205093, #2146343, #2134274, #2441832 and CDC-RFA-FT-23-0069.

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

# A. Algorithm Tables

The CoDD with block diffusion algorithm is presented in Algorithm 1 and the one with full diffusion is illustrated in Algorithm 2.

---

**Algorithm 1** CoDD with Block Diffusion

---

**Require:** Distributions $p_{\boldsymbol{\omega}}(\boldsymbol{x}_0)$ and $p_{\boldsymbol{\theta}}(\boldsymbol{x}_0)$ parameterized by the PC and the Transformer, respectively; threshold $\gamma$; temperature $\tau$; sequence length $L$; block size $L_{\mathrm{b}}$; number of decoding steps $n$

1: Initialize sequence $\boldsymbol{x} \leftarrow$ array of size $L$ filled with `<MASK>`
2: **for** $k = 0$ **to** $L/L_{\mathrm{b}} - 1$ **do**
3:     Define block window: `ids` $\leftarrow [k \cdot L_{\mathrm{b}}, (k+1) \cdot L_{\mathrm{b}}]$
4:     Initialize current block state $\boldsymbol{x}_n[\texttt{ids}]$ with `<MASK>`
5:     **for** $t = n$ **to** $1$ **do**
6:         Compute mask ratio $r_m$ within the current block
7:         Get Transformer outputs: $\boldsymbol{\theta} \leftarrow f_{\boldsymbol{\varphi}}(\boldsymbol{x}_t)$
8:         **if** $r_m < \gamma$ **then**
9:            # Adaptive Activation (Section 5.3)
10:            Sample $\hat{\boldsymbol{x}}_0$ from $p_{\boldsymbol{\theta}}^{1/\tau}(\mathbf{X}_0) \cdot p_{\boldsymbol{\omega}}(\mathbf{X}_0)$
11:         **else**
12:            Sample $\hat{\boldsymbol{x}}_0$ from $p_{\boldsymbol{\theta}}^{1/\tau}(\mathbf{X}_0)$
13:         **end if**
14:         Re-mask to next state: $\boldsymbol{x}_{t-1}[\mathrm{idx}] \sim q(\cdot \mid \hat{\boldsymbol{x}}_0, \boldsymbol{x}_t[\mathrm{idx}])$       # Remask each token w.p. $\alpha_t$
15:     **end for**
16:     $\boldsymbol{x}[\mathrm{idx}] \leftarrow \boldsymbol{x}_0[\mathrm{idx}]$    # Commit generated block to context
17: **end for**
18: **Return:** $\boldsymbol{x}$

---

**Algorithm 2** CoDD with Full Diffusion

---

**Require:** Distributions $p_{\boldsymbol{\omega}}(\boldsymbol{x}_0)$ and $p_{\boldsymbol{\theta}}(\boldsymbol{x}_0)$ parameterized by the PC and the Transformer; threshold $\gamma$; temperature $\tau$; sequence length $L$; PC window size $W$; number of decoding steps $n$

1: Initialize sequence $\boldsymbol{x}_n \leftarrow$ array of size $L$ filled with `<MASK>`
2: **for** $t = n$ **to** $1$ **do**
3:     Compute global mask ratio $r_m$
4:     Get Transformer outputs: $\boldsymbol{\theta} \leftarrow f_{\boldsymbol{\varphi}}(\boldsymbol{x}_t)$
5:     # Adaptive Activation with Dynamic Windowing
6:     Identify target indices $\mathcal{M}$ to be decoded following the baseline's strategy
7:     Select window $[u, v]$ of size $W$ that covers the largest number of indices in $\mathcal{M}$
8:     **if** $r_m < \gamma$ **and** $|\mathcal{M}_{\mathrm{win}}| > 1$ **then**
9:         Sample $\hat{\boldsymbol{x}}_0^{u:v}$ from $p_{\boldsymbol{\theta}}^{1/\tau}(\mathbf{X}_0^{u:v}) \cdot p_{\boldsymbol{\omega}}(\mathbf{X}_0^{u:v})$ # PC-guided joint sampling
10:     **else**
11:         Sample $\hat{\boldsymbol{x}}_0^{u:v}$ from $p_{\boldsymbol{\theta}}^{1/\tau}(\mathbf{X}_0^{u:v})$
12:     **end if**
13:     Re-mask to next state: $x_{t-1}^i \sim q(\cdot \mid \hat{x}_0^i, x_t^i)$ if $i \in [u, v]$; otherwise $x_{t-1}^i \leftarrow x_t^i$
14: **end for**
15: **Return:** $\boldsymbol{x}_0$

---

## B. Experiment details

For LLaDA-Instruct-8B, we use greedy decoding, while for Dream-Instruct-7B, we adopt the default sampling hyperparameters, with temperature $0.2$ and top-$p$ $0.95$.

We set the maximum generation length to 512 tokens for all tasks, in practice, models typically emit the `<eos>` token before reaching this limit, at which point we terminate decoding.

For GSM8K and MBPP, we use the standard system prompt, *"You are a helpful assistant"*. For GPQA and MATH500, we modify the system prompt to elicit reasoning and explicitly specify required answer format, following Israel et al. (2025)'s protocol.

**Ablation Studies on $\gamma$ and $\tau$.** In Tables 5 and 6, we show the performance of CoDD (LLaDA) with varying hyperparameters $\gamma$ (the adaptive activation threshold introduced in Sec. 5.3) and $\tau$ (the sampling temperature).

*Table 5.* **Performance Comparison (GPQA) on LLaDA Model with Block Diffusion.** Accuracy (%) across 256, 128, and 64 diffusion steps with varying `pc frac` (i.e., $\gamma$) and `pc temp` (i.e., $\tau$). **Bold** indicates the best performance for that step count.

| PC Fraction | 256 Steps | | | | 128 Steps | | | | 64 Steps | | | |
|---|---|---|---|---|---|---|---|---|---|---|---|---|
| | Random | | Low Conf | | Random | | Low Conf | | Random | | Low Conf | |
| | 0.1 | 0.2 | 0.1 | 0.2 | 0.1 | 0.2 | 0.1 | 0.2 | 0.1 | 0.2 | 0.1 | 0.2 |
| 0.3 | 24.33 | 21.65 | 19.42 | 21.43 | **21.65** | 17.41 | – | – | 10.04 | 10.49 | **9.15** | 8.93 |
| 0.5 | 24.55 | 24.33 | 23.66 | **24.55** | 19.20 | 18.75 | 16.52 | 16.52 | 10.04 | 10.49 | **9.15** | 8.93 |
| 0.6 | 19.64 | **26.34** | 24.11 | 22.77 | 19.42 | 17.41 | 17.63 | 16.07 | 11.16 | **11.38** | 8.93 | 8.48 |
| 0.7 | 21.88 | 22.10 | – | – | 18.97 | **19.64** | 16.74 | 15.40 | 11.16 | **11.38** | 8.93 | 8.48 |
| 0.8 | – | – | – | – | – | – | 17.86 | 15.18 | – | – | – | – |

*Table 6.* **Performance Comparison (MBPP) on Dream Model with Block Diffusion.** Accuracy (%) across 256, 128, and 64 diffusion steps with varying `pc frac` (i.e., $\gamma$) and `pc temp` (i.e., $\tau$). **Bold** indicates the best performance for that step count.

| PC Fraction | 256 Steps | | | | 128 Steps | | | | 64 Steps | | | |
|---|---|---|---|---|---|---|---|---|---|---|---|---|
| | Random | | Low Conf | | Random | | Low Conf | | Random | | Low Conf | |
| | 0.1 | 0.2 | 0.1 | 0.2 | 0.1 | 0.2 | 0.1 | 0.2 | 0.1 | 0.2 | 0.1 | 0.2 |
| 0.3 | 27.80 | 28.00 | 46.80 | **47.20** | 16.60 | 16.40 | 30.40 | 30.60 | 5.00 | 4.80 | **20.20** | 19.60 |
| 0.5 | 27.40 | 27.00 | 46.80 | **47.20** | 17.20 | 16.80 | 30.60 | 31.00 | 5.00 | 4.80 | **20.20** | 19.60 |
| 0.7 | 28.00 | 27.40 | 46.00 | 45.00 | 17.40 | 16.00 | 29.60 | **31.60** | 5.00 | 4.80 | 19.40 | **20.20** |

**Additional Results on Dream.** Table 7 presents additional empirical results of CoDD on the Dream model.

*Table 7.* **Performance Comparison (Dream) with Block Diffusion.** Accuracy (%) of Dream baselines versus their CoDD-augmented versions.

| Model | Decoding Strategy / Diffusion Steps | MATH500 | | | GSM8K | | | GPQA | | | MBPP | | |
|---|---|---|---|---|---|---|---|---|---|---|---|---|---|
| | | 256 | 128 | 64 | 256 | 128 | 64 | 256 | 128 | 64 | 256 | 128 | 64 |
| **Dream** | Random | 22.20 | 10.00 | 2.40 | 47.69 | 33.97 | 14.10 | 23.21 | 11.83 | 8.26 | 27.80 | 16.80 | 5.20 |
| | Margin | 37.20 | 19.40 | 2.00 | 74.98 | 58.53 | 24.26 | 23.66 | 13.17 | 5.13 | 44.40 | 29.20 | 18.00 |
| | Low Confidence | **45.00** | 19.60 | **3.60** | **75.44** | 59.51 | 21.99 | 25.22 | 11.83 | 4.46 | 44.40 | 26.20 | 16.00 |
| | Entropy | 38.40 | 20.20 | 2.40 | 75.06 | 56.86 | 24.03 | 20.76 | 12.05 | 5.13 | 44.60 | 29.00 | 18.00 |
| **CoDD** | Random | 24.80 | 10.00 | 2.40 | 48.90 | 34.27 | **31.70** | 24.11 | 12.72 | **8.48** | 28.00 | 17.40 | 5.00 |
| | $\Delta$ *performance* | +2.60 | 0.00 | 0.00 | +1.21 | +0.30 | +17.60 | +0.89 | +0.89 | +0.22 | +0.20 | +0.60 | -0.20 |
| | Entropy | 44.40 | **21.80** | 3.40 | 75.20 | 57.16 | 24.94 | **25.89** | **13.39** | 4.91 | **47.20** | **31.60** | **20.20** |
| | $\Delta$ *performance* | +6.00 | +1.60 | +1.00 | +0.14 | +0.30 | +0.91 | +5.13 | +1.34 | -0.22 | +2.60 | +2.60 | +2.20 |

# C. Additional Details on Probabilistic Circuits

This section provides the formal definition of decomposability and introduce how to sample from a PC. We start with the definition of decomposability.

**Definition C.1** (Decomposability). Let the scope $\phi(n)$ denote the collection of variables associated with the leaf nodes descending from node $n$. A PC is decomposable if the children of any product node have disjoint scopes ($\forall c_1, c_2 \in \text{ch}(n)$ s.t. $c_1 \neq c_2, \phi(c_1) \cap \phi(c_2) = \varnothing$).

Intuitively, decomposability requires every product node of the PC to model "valid factorized distributions", ensuring that no variable is multiplied by itself. This property is crucial for the efficient computation of the partition function and marginals.

We now detail the algorithms for sampling from a decomposable PC.

**Unconditional Sampling.** Sampling a complete configuration $\mathbf{x} \sim p_\omega(\mathbf{x})$ is performed via a single top-down pass starting from the root node $n_r$. The process proceeds recursively:

- **Sum Node:** If the current node $n$ is a sum node, we sample a child $c \in \text{ch}(n)$ according to the categorical distribution defined by the edge weights $\{\omega_{n,c}\}_{c \in \text{ch}(n)}$.

- **Product Node:** If $n$ is a product node, we recurse to *all* children $c \in \text{ch}(n)$ independently. This is valid because decomposability ensures the children model disjoint sets of variables.

- **Leaf Node:** If $n$ is a leaf node, we simply sample a value from its univariate distribution $g_n(\cdot)$.

**Conditional Sampling.** To sample from the conditional distribution $p_\omega(\boldsymbol{x}_{\text{miss}} \mid \boldsymbol{x}_{\text{obs}})$ given some evidence (e.g., unmasked tokens), we execute one bottom-up evaluation pass followed by one top-down sampling pass.

1. **Forward Pass (Bottom-Up):** We first compute the likelihood of the evidence at every node. For leaf nodes, we evaluate the probability of the observed value (or 1 if the variable is missing). These values are propagated upward to the root, where node $n$ computes the likelihood of the evidence restricted to its scope, denoted as $L_n(\mathbf{x}_{\text{obs}})$.

2. **Backward Pass (Top-Down):** We sample strictly from the missing variables $\mathbf{x}_{\text{miss}}$ using a logic similar to the unconditional case, but with "evidence-adjusted" weights:

   - At a **Sum Node** $n$, instead of using the raw parameters $\omega_{n,c}$, we sample a child $c$ using the posterior probability given the evidence:
     $$p(c \mid n, \mathbf{x}_{\text{obs}}) \propto \omega_{n,c} \cdot L_c(\mathbf{x}_{\text{obs}}).$$
   - At a **Product Node**, we recurse to all children as before.
   - At a **Leaf Node**, if the variable is observed, we fix it to the observed value; otherwise, we sample from the leaf distribution.

# D. Justification of the Partition Function Computation Algorithm

In this section, we formally justify the efficiency of our algorithm for computing the partition function $Z$ of the joint product distribution $\hat{p}_{\theta,\omega}(\boldsymbol{x}) \propto p_\omega(\boldsymbol{x}) \cdot p_\theta(\boldsymbol{x})$. We show that this calculation is mathematically equivalent to computing the marginal probability of a Probabilistic Circuit (PC) under *independent virtual evidence*, a query known to be tractable for decomposable circuits. We first define the general form of a virtual-evidence query.

**Definition D.1** (Independent Virtual-Evidence). Let $\mathbf{X} = \{X_1, \ldots, X_L\}$ be a set of discrete random variables. A set of functions $\mathcal{W} = \{w_1, \ldots, w_L\}$ is called *independent virtual evidence* if each $w_i$ is a univariate non-negative function mapping the domain of $X_i$ to $\mathbb{R}_{\geq 0}$ (i.e., $w_i : \text{Dom}(X_i) \rightarrow \mathbb{R}_{\geq 0}$).

The probability of observing this virtual evidence under a distribution $p(X)$ is defined as the expected product of these weights:

$$P(\mathcal{W}) := \sum_{\boldsymbol{x} \in \text{Dom}(\mathbf{X})} p(\boldsymbol{x}) \prod_{i=1}^{L} w_i(x_i). \tag{8}$$

**Mapping to the partition function in Equation (5).** Recall that in our framework, the joint distribution is defined as the product of a PC prior $p_{\boldsymbol{\omega}}(\boldsymbol{x}_0)$ and a fully factorized conditional term $p_{\boldsymbol{\theta}}(\boldsymbol{x}_0 \mid \boldsymbol{x}_t) = \prod_{i=1}^{L} p_{\boldsymbol{\theta}}(x_0^i \mid \boldsymbol{x}_t)$. The partition function $Z$ is the normalization constant required to make this product a valid distribution:

$$Z = \sum_{\boldsymbol{x}_0} p_{\boldsymbol{\omega}}(\boldsymbol{x}_0) \cdot p_{\theta}(\boldsymbol{x}_0 \mid \boldsymbol{x}_t) = \sum_{\boldsymbol{x}_0} p_{\boldsymbol{\omega}}(\boldsymbol{x}_0) \prod_{i=1}^{L} p_{\theta}(x_0^i \mid \boldsymbol{x}_t).$$

By defining the weight functions as $w_i(x_0^i) := p_{\theta}(x_0^i \mid \boldsymbol{x}_t)$, we observe that computing $Z$ is exactly equivalent to computing the probability of soft evidence $P(\mathcal{W})$ (Eq. (8)) where the "evidence" is provided by the neural network's factorized logits.

**Tractability.** To establish the efficiency of this computation, we reuse Theorem 1 from Liu et al. (2024).

**Theorem D.2.** *For any smooth and decomposable PC $p_{\omega}$ over variables $X$, and any set of independent soft evidence $\mathcal{W}$, the quantity $P(\mathcal{W})$ can be computed exactly in time linear in the size of the circuit.*

# E. Alternative Ways to Instantiate $\hat{p}_{\boldsymbol{\theta},\boldsymbol{\omega}}$

In the main text (Sec. 5.3), we observed that the dependency structure of natural language evolves significantly throughout the diffusion process. Our current implementation uses a single static PC $p_{\boldsymbol{\omega}}(\boldsymbol{x}_0)$, which essentially learns the "average" dependency structure across all timesteps. While our adaptive activation threshold $\gamma$ mitigates this by deactivating the PC when it is likely to be misspecified, a more rigorous solution is to allow the neural backbone to dynamically modulate the structural prior itself. In this section, we outline two concrete architectural extensions to achieve this noise-conditional structural guidance. We consider them as promising directions for future work.

## E.1. Latent-Space Modulation

The first approach leverages the interpretation of PCs as deep latent variable models (Peharz et al., 2016; Liu et al., 2023). A PC can be viewed as a mixture model over a set of discrete latent variables $\mathbf{z}$, where each instantiation of $\mathbf{z}$ corresponds to a specific "routing path" through the sum nodes of the circuit. The joint distribution is given by $p_{\boldsymbol{\omega}}(\boldsymbol{x}_0) = \sum_{\mathbf{z}} p_{\boldsymbol{\omega}}(\boldsymbol{x}_0, \mathbf{z})$. Currently, the Transformer backbone $p_{\theta}$ only outputs potentials over the observed variables $\boldsymbol{x}_0$. We propose augmenting the Transformer to output a factorized distribution over *both* the data and the PC's latent variables:

$$p_{\boldsymbol{\theta}}(\boldsymbol{x}_0, \mathbf{z} \mid \boldsymbol{x}_t) = p_{\boldsymbol{\theta}}(\boldsymbol{x}_0 \mid \boldsymbol{x}_t) \cdot p_{\boldsymbol{\theta}}(\mathbf{z} \mid \boldsymbol{x}_t). \tag{9}$$

The hybrid joint distribution then becomes:

$$\hat{p}_{\boldsymbol{\theta},\boldsymbol{\omega}}(\boldsymbol{x}_0 \mid \boldsymbol{x}_t) \propto \sum_{\mathbf{z}} p_{\boldsymbol{\omega}}(\boldsymbol{x}_0, \mathbf{z}) \cdot p_{\boldsymbol{\theta}}(\boldsymbol{x}_0, \mathbf{z} \mid \boldsymbol{x}_t). \tag{10}$$

**Intuition.** In this formulation, the Transformer does not just predict *what* tokens are likely ($p_{\theta}(\boldsymbol{x}_0)$), but also *which dependency mode* is currently active ($p_{\theta}(\mathbf{z})$). For instance, if the PC contains distinct sub-circuits for "text generation" and "arithmetic reasoning", the Transformer can output high probabilities for the latent variables associated with the relevant sub-circuit based on the current context $x_t$. This effectively "steers" the PC, upweighting the relevant structural components and downweighting irrelevant ones.

## E.2. Parameter-Space Modulation

The second approach is to have the neural network directly predict the parameters of the structural prior. Instead of learning a fixed set of weights $\omega$, we can treat a subset of the PC parameters as a function of the context $\boldsymbol{x}_t$, and model it by the Transformer backbone (similar to $\boldsymbol{\theta}$). This provides additional flexibility to steer the PC for distinct contexts.

# F. Qualitative Examples

We provide a qualitative comparison between the baseline LLaDA model and CoDD on the MATH500 dataset under a constrained compute budget of 64 diffusion steps.

---

**Baseline (LLaDA) – 64 Steps**

---

**Question:** If $2^8 = 4^x$, what is the value of $x$?

**Generation:**
To solve the equation $2^8 = 4^x$, we need to express both sides of thex in the in the same base..
We know that
(4222222222222222222222222222222222222222222222222222222222222222222... *[repetition]*

**Status: Incorrect (Degeneration)**

---

**CoDD (Ours) – 64 Steps**

---

**Question:** If $2^8 = 4^x$, what is the value of $x$?

**Generation:**
To solve the equation $2^8 = 4^x$, we need to express both sides of the same base. Let's follow these steps:
1. **Express 4 as a power of 2:**
$$4 = 2^2$$

2. **Rewrite the $4^x$ using the expression of 2:**
$$4^x = (2^2)^x$$

3. **Simplify the right-hand side:**
$$(2^2)^x = 2^{2x}$$

4. **Set the exponents equal:**
$$2^8 = 2^{2x} \implies 8 = 2x$$

5. **Solve for $x$:**
$$x = 4$$

Therefore, the value of $x$ is $\boxed{4}$.

**Status: Correct**

---

*Figure 5.* **Qualitative comparison: Exponent Equation (64 steps).** At reduced sampling budgets, the baseline model suffers from severe mode collapse (repeating tokens). CoDD effectively steers the generation to recover coherent reasoning.

---

**Baseline (LLaDA) – 64 Steps**

---

**Question:** Expand and simplify: $x(x(1 + x) + 2x) - 3(x^2 - x + 2)$

**Generation:**
To expand and simplify the expression... we will follow these steps:
1. **Expand the expression inside the parentheses:**

$$x(1 + x) + 2x = x^2 + 3x$$

... *[steps 2-5 omitted for brevity]* ...
5. **Combine the expanded terms:**
$$x^3 + 3x^2 - 3x^2 + 3x - 6$$

6. **Combine like terms:**

$$x^3 + 3x^2 - 3x^2 + 3x - 6 = x^3$$

Therefore, the simplified expression is:

$$\boxed{x^3}$$

**Status: Incorrect (Arithmetic Hallucination)**

---

**CoDD (Ours) – 64 Steps**

---

**Question:** Expand and simplify: $x(x(1 + x) + 2x) - 3(x^2 - x + 2)$

**Generation:**
To expand and simplify... we will follow these steps:
1. **Expand the expression inside the parentheses:**

$$x(1 + x) = x + x^2 \implies x + x^2 + 2x = x^2 + 3x$$

2. **Multiply by the outer x:**
$$x(x^2 + 3x) = x^3 + 3x^2$$

3. **Distribute the -3:**
$$-3(x^2 - x + 2) = -3x^2 + 3x - 6$$

4. **Combine and simplify:**
$$x^3 + 3x^2 - 3x^2 + 3x - 6 = x^3 + 3x - 6$$

Therefore, the expanded form is:
$$\boxed{x^3 + 3x - 6}$$

**Status: Correct**

*Figure 6.* **Algebraic simplification at low compute (64 steps).** The baseline model correctly derives the intermediate terms but fails at the final combination step, effectively "forgetting" the remaining terms ($3x - 6$). CoDD maintains coherence through the multi-step reasoning to reach the correct solution.

