# OpenReview forum: "Breaking the Factorization Barrier in Diffusion Language Models"
_ICML.cc/2026/Conference — ICML 2026 regular_

### Official Review · Reviewer_xM9V · 2026-03-10

**Soundness:** 3
**Presentation:** 3
**Significance:** 4
**Originality:** 3
**Overall Recommendation:** 5
**Confidence:** 4

**Summary:**

This paper proposes a framework called CODD, that focuses on the problem that, during inference, parallel generation in dLLMs relies on the assumption that simultaneously predicted tokens are independent (factorization of the marginals across positions). This makes the generated samples suffer from incoherences, especially in few-steps regimes.

The core method introduced by the paper is to add a lightweight layer to the output to mitigate this factorization assumption. This is done with a Probabilistic Circuits (PC) layer, that is a class of deep network introduced in previous works.

On the experimental side, CODD is used as a post training method applied on existing dLLMs base-models, and is compared against RL methods in terms of quality and efficiency.

**Compliance With Llm Reviewing Policy:**

Affirmed.

**Final Justification:**

My main questions are now resolved. Since I also think the paper addresses a genuinely important and non trivial problem for diffusion language models, I am now clearly in favor of acceptance. I am therefore increasing my score to support the paper toward acceptance.

**Key Questions For Authors:**

1) I would have appreciated a clearer comparison with approaches such as Discrete Copula Diffusion (DCD) [1]. In particular, why introducing Probabilistic Circuits, which seem to be a fairly nontrivial additional modeling tool, is preferable to using an autoregressive copula-style model as in DCD? Related to this, the paper states around line 418 that “the use of such additional models incurs significant computational overhead” but this is not entirely convincing to me in the experimental setting considered here, since the method is mostly presented as a post-training procedure and compared against RL methods. A more explicit conceptual and computational comparison with DCD would strengthen the positioning and impact of the paper.

2) I also had a theoretical question regarding the objective used in Equation (7). Standard ELBO-based objectives in diffusion language models come from maximizing a lower bound on the model likelihood. Here, as explained in line 242, the paper seems to replace $p_{\varphi}$ with $p_{\theta,\omega}$ in a rather direct way. Is there any theoretical justification for this substitution, or any evidence that the resulting objective still has a principled likelihood-based interpretation? This may be a simple point, but it is not clear to me and I think clarifying it would strengthen the paper.

3) In the experiments, the method is mainly used as a fine-tuning or post-training procedure on top of pretrained models. This made me wonder why the authors do not also consider training from scratch, as suggested in line 102 of the paper "the inference layer can be optimized either jointly with the neural backbone or separately as a lightweight, plug-and-play module." It is not clear to me whether this was attempted and found ineffective, or whether it was simply outside the intended scope of the paper. My intuition is that direct joint training may be difficult, especially if the modification amounts to plugging $p_{\theta,\omega}$ into the standard MDM training loss, but if that is indeed the case, I think the paper would benefit from discussing it explicitly. Otherwise, the contribution may be better framed primarily as a highly efficient post-training alternative to heavier RL-based pipelines.

4) The paper states that the framework is agnostic to the underlying diffusion strategy, and discusses both block diffusion and full diffusion. How far this generality really extends beyond masking-based settings. More precisely, does the method depend in a deeper way on the type of forward process, or is it genuinely applicable to a broader family of discrete diffusion processes?

5) Since I am not very familiar with Probabilistic Circuits, I felt that the appendix could have provided more background on them. The level of explanation in the main text is sufficient to follow the overall method, but I would have expected a somewhat deeper introduction or supplementary discussion in the appendix. Currently, appendix C and D are minimal. This is not a critical issue, but I think it would make the paper more accessible to a broader audience.

6) Minor issue ; I found the notation between uppercase $\mathbf{X}$ and lowercase $\mathbf{x}$ somewhat confusing. Can you explain why do we switch from one to an other? I assume this is meant to distinguish random variables from their realizations but I am not sure.

Despite all these questions, I think that the paper is good overall, and is one of the rare paper that try to tackle this important problem. So I am slightly in favor for acceptance, and I am open to augment my score further if my questions are solved.

[1] Discrete copula diffusion, Liu et al. 2025

**Limitations:**

A discussion about limitation is not provided. I suggest the missing discussion of the results obtained against DCD can be part of this discussion.

**Strengths And Weaknesses:**

- Strenghts ;

  - The paper treats a really important problem in the area of dLLMs, about their promises of parallel generation.
  - Empirical results are technically sounds ; CODD consistently improves compared to a based dLLMs without the additional layer. CODD is really efficient, in terms of training time when compared to heavy RL baseline for fine-tuning.
  - The paper is well written overall, and it's quite clear to understand the backgrounds and the problem it aims to tackle.

- Weaknesses ;

  - CODD is only applied for fine-tuning tasks in the manuscript.
  - Missing baseline comparison : The pros and the cons of CODD against [1] (a baseline, cited in the paper, that tackles the same problem) are not studied.

[1] Discrete copula diffusion, Liu et al. 2025

---

> ### Author Rebuttal · Authors · 2026-03-31
>
> We thank the reviewer for their thoughtful and detailed feedback, and for recognizing the soundness and significance of our work.
>
> > W1/Q3: CODD is only applied for fine-tuning tasks; why not train from scratch?
>
> While our experiments focus on fine-tuning, CoDD is a general architectural framework that is not limited to fine-tuning tasks. Specifically, we can optimize the model end-to-end and apply gradient-based updates to both the Transformer backbone and the PC when training from scratch. Further the objective (Eq. 7) can be directly used when training from scratch.
>
> For the fine-tuning case, CoDD yields massive performance gains in just ~3 GPU hours of training (compared to ~150 hours for diff-GRPO). As demonstrated, this highly efficient fine-tuning adaptation already outperforms resource-intensive diffu-GRPO in reasoning tasks like MATH500.
>
> We agree that scaling CoDD to full pre-training from scratch is a highly promising future direction. However, given the computational and engineering costs, we focused this work on fine-tuning tasks to provide the community with an immediate, lightweight plug-and-play solution.
>
> > W2/Q1: Comparison with DCD.
>
> Following DCD, we utilized SEDD as the base diffusion model. We trained the PC using samples generated by GPT-2, and evaluated both methods on the WikiText-103 validation set. For each instance, we compute the LL of the true completion given a randomly truncated context (Ctx = context length in tokens).
> |Ctx|SEDD|+DCD|+CoDD|
> |---|---|---|---|
> |[20,40)|-14.09|-4.00|-2.96|
> |[40,60)|-14.03|-3.94|-2.92|
> |[60,80)|-14.00|-3.93|-2.91|
> |[80,100)|-13.79|-3.74|-2.78|
> |[100,120)|-13.60|-3.57|-2.75|
> |[120,140)|-13.64|-3.62|-2.71|
> |[140,160)|-13.74|-3.79|-2.85|
> |[160,180)|-13.93|-3.82|-2.98|
> |[180,200)|-13.63|-3.55|-2.66|
> |[200,220)|-14.12|-4.01|-2.75|
> |[220,240)|-13.15|-3.01|-2.39|
> |[240,260)|-13.43|-3.07|-2.30|
>
> The empirical results demonstrate that CoDD consistently achieves higher LL than DCD across all context lengths.
> CoDD and DCD both aim to capture joint token dependencies, but PCs offer key advantages for diffusion settings.
>
> DCD's autoregressive (AR) copula assumes a fixed ordering (its non-AR variant is very slow compared to the AR version), whereas discrete diffusion requires modeling dependencies under arbitrary and dynamic masking patterns. PCs are uniquely suited for this because they support exact marginalization for any subset of missing tokens in a single forward pass, whereas an autoregressive copula would require sequential re-ordering or approximation.
> Computationally, querying an autoregressive copula for $K$ tokens at each diffusion step introduces a $O(K)$ sequential bottleneck; Our PC performs inference in $O(1)$ sequential time via a parallelizable bottom-up pass.
> > Q2: Theoretical justification for substituting $p_\phi$ with $p_{\theta, \omega}$ in Eq. (7).
>
> Yes, there is a direct principled justification. The standard dLLM ELBO introduces the factorization at the step where the denoising distribution $ p_\phi(x_0 | x_t)$ is approximated as a product of independent marginals $\prod_i p_\phi(x^i_0 | x_t)$. We substitute this with the joint distribution $p_{\theta, \omega}(x_0 | x_t) := \frac{1}{Z(x_t)} \cdot p_\omega(x_0 | x_t) \cdot p_\theta(x_0)$ at exactly this step. The resulting objective (Eq. 7) is therefore the same ELBO derivation without the independence assumption, and retains a valid evidence lower-bound interpretation under the joint model. We will add a self-contained derivation in the appendix.
>
> > Q4: Does CoDD generalize beyond masking-based diffusion?
>
> For alternative discrete diffusion strategies, consider: (1) Top-k conditioned models (next-step context is defined by the most confident token predictions); CoDD applies directly as the PC can condition on "soft evidence" such as p(y|x = "cat" or x = "dog"). (2) Token-editing models (where tokens can be changed rather than only unmasked); CoDD also applies without modification, as the product distribution is defined over all positions regardless of the edit type. (3) Absorbing diffusion with different absorbing states; the PC operates on the denoised token space and is agnostic to the specific noise injection mechanism. In all cases, replacing the factorized output with $p_{\theta,\omega}$ requires no changes to the diffusion schedule or sampling procedure.
>
> > Q5: Details on PCs.
>
> We appreciate the suggestion and we will expand Appendix C with a self-contained introduction to PCs including structural properties, algorithmic passes, and inference mechanics.
>
> > Q6: Notation between uppercase and lowercase.
>
> This follows the standard convention: uppercase $X$ denotes a random variable, and lowercase $x$ denotes a specific realization or sample. We will add an explicit clarification of this notation at the start of Section 2.
>
> We hope our response has addressed your questions, and would be grateful if you could reconsider your score in light of the new results and clarifications.

---

> > ### Author Rebuttal · Reviewer_xM9V · 2026-04-02
> >
> > Thank you for the detailed rebuttal and for addressing my questions.
> >
> > The additional comparison with DCD was especially helpful, since this was one of my main concerns. I think the new results and discussion substantially improve the positioning of the paper. In particular, the explanation of why PCs are better suited than AR-copula-style models for arbitrary masking patterns in diffusion is convincing, and the empirical comparison makes the tradeoff much clearer. I strongly encourage the authors to add that experiment to the paper.
> >
> > I also appreciate the clarification regarding Eq. (7). The explanation that the objective follows the usual dLLM ELBO derivation, with the factorized family replaced by the joint model $\hat p_{\theta,\omega}$, addresses my concern at a conceptual level. Assuming the promised derivation is added clearly in the appendix, I consider this point resolved.
> >
> > Regarding training from scratch, I now better understand that the method is intended as a general framework, while the experiments focus on the lightweight post-training regime because this already provides a strong and very compute-efficient use case. But to me it is still not trivial to see how it is possible to train from scratch with the current framework. I still think the current paper most convincingly validates CoDD as a plug-and-play post-training method rather than as a fully demonstrated end-to-end training framework, but I am satisfied with the clarification.
> >
> > The answers regarding applicability beyond masking-based diffusion, the planned appendix expansion on PCs, and the notation issue were also helpful.
> >
> > Overall, my main questions are now resolved. Since I also think the paper addresses a genuinely important and non trivial problem for diffusion language models, I am now clearly in favor of acceptance. I am therefore increasing my score to support the paper toward acceptance.

---

> > > ### Author Response · Authors · 2026-04-03
> > >
> > > Thank you so much for your positive feedback! We are encouraged that our rebuttal and the additional experiments addressed your concerns.
> > >
> > > We fully agree that the comparison with DCD highlights the unique advantages of PCs and improves the positioning of our paper. We will incorporate these results and the accompanying discussion into the revised version.
> > >
> > > We also agree that training from scratch within the current CoDD framework introduces non-trivial challenges that are not fully explored in the current experiments. In the revised version, we will explicitly clarify the scope of our paper to emphasize CoDD as a plug-and-play post-training method. We will add a discussion acknowledging that while the framework theoretically supports end-to-end training, fully validating this at scale remains an exciting direction for future work.
> > >
> > > We also reaffirm that the promised derivation of Eq. (7), the expanded details on Probabilistic Circuits, and the notation fixes will be integrated into the appendix and main body.
> > >
> > > Thank you again for your insightful review. Your feedback has been extremely helpful in refining the positioning and clarity of our work, and we are grateful for your recognition of its importance to the diffusion language modeling community.

---

### Official Review · Reviewer_9C6n · 2026-03-10

**Soundness:** 3
**Presentation:** 3
**Significance:** 2
**Originality:** 3
**Overall Recommendation:** 5
**Confidence:** 3

**Summary:**

This paper aims at improving diffusion language models. It tries to circumvent the trade-off between accuracy (which is maximized when only a single token is reconstructed at a time) and latency (which is maximized when many token positions are reconstructed at once, and usually independently, which prevents good accuracy by ignoring dependencies even when the model would otherwise be able capture them, if sampling one token at a time).

**Compliance With Llm Reviewing Policy:**

Affirmed.

**Final Justification:**

In light of the answers, I have raised my scores.

**Key Questions For Authors:**

The PC looks like a hierarchical mixture where not all variables are involved in each component at each level of the hierarchy. What is the evidence that this kind of architecture generalizes as well as the current neural net state-of-the-art?

If I understand well, the neural net is limited to make the factorized reconstructions, not to directly capture the interactions explicitly?

See my question about actually reporting latency-vs-accuracy curves.

**Limitations:**

None discussed.

**Strengths And Weaknesses:**

Soundness and originality

The approach seems sound and combines existing methods to address a problem that is quite relevant with modern diffusion language models.

Presentation

There seems to be a notation problem in (5): on the LHS we have x_t but it is abent on the RHS. Is it that p_w and p_theta are implicitly functions of x_t?

Significance

What I would like to see are comparative curves (across methods) showing total compute or total latency vs entropy (or other appropriate goodness of generation metrics), because right now it is not clear if the overhead in compute or latency is compensated by the improvement in entropy and if the gain is significant compared with the differences between methods in general.

---

> ### Author Rebuttal · Authors · 2026-03-31
>
> We thank the reviewer for recognizing the soundness and significance of our approach.
>
> > W1: Notation issue in Eq. (5).
>
> Thank the reviewer for spotting the typo. On the RHS, $p_\theta$ uses an implicit condition as the neural network does not explicitly compute conditional probability, but $p_\omega(x_0|x_t)$ should be explicitly conditioned since the PC performs exact probabilistic inference. The correct notation should be:
>
> $p_{\theta, \omega}(x_0 | x_t) := \frac{1}{Z(x_t)} \cdot p_\omega(x_0 | x_t) \cdot p_\theta(x_0)$.
>
> We will fix this in the revision.
>
> > Q1: What evidence that the PC architecture generalizes as well as neural net state-of-the-art?
>
> The comparison is not about replacing neural networks, but rather solving a structural limitation they fundamentally cannot overcome. When generating multiple tokens simultaneously, existing diffusion LLMs are mathematically constrained to output independent, factorized distributions. No matter how advanced or large the neural network is, it physically cannot capture the joint dependencies between these co-sampled tokens. The PC architecture specifically solves this "factorization barrier" by explicitly modeling the local joint distribution, doing exactly what the neural network cannot.
>
> Empirically, CoDD consistently improves performance over strong factorized baselines across four benchmarks, two architectures, and four decoding strategies (Tables 1 and 2). Specifically, on MATH500, CoDD maintains higher accuracy compared to standard RL-finetuning methods (d-GRPO), proving that the PC's hierarchical structure successfully "fills in" the dependency gaps that cause standard models to fail in multi-token prediction.
>
> > Q2: The neural net is limited to factorized reconstructions and cannot capture interactions directly?
>
> Yes, as mentioned in Section 3, the standard neural network backbone is structurally restricted to factorized reconstructions because explicitly parameterizing a joint distribution over a large vocabulary would require a prohibitively large output layer, which is intractable for any realistic setting. CoDD uses PC to capture dependency while not exploding the computational complexity.
>
> > W2/Q3: Latency-vs-accuracy curves to show the overhead is justified.
>
> We thank the reviewer for this excellent suggestion. We have compiled the comparative results in the table below and will include a Pareto frontier curve (Latency vs. Accuracy) in the revised manuscript.
>
> | Diffusion Steps | Method | Accuracy (%) | Latency (sec/sample) |
> |----------------|--------|-------------|---------------------|
> | **256 Steps** | LLaDA | 36.0% | 11.51 |
> | | diffu-GRPO | 39.8% | 15.31 |
> | | CoDD  | 41.0% | 12.11 |
> | **128 Steps** | LLaDA | 31.6% | 5.49 |
> | | diffu-GRPO | 32.4% | 7.64 |
> | | CoDD | 33.8% | 6.18 |
> | **64 Steps** | LLaDA | 12.6% | 2.86 |
> | | diffu-GRPO | 12.8% | 3.81 |
> | | CoDD | 15.8% | 3.04 |
>
> As demonstrated in the table above, CoDD consistently outperforms diffu-GRPO in both generation quality and computational efficiency. At 64 steps, CoDD adds only 6.3% latency over LLaDA while improving accuracy by 3.2 points; diffu-GRPO adds 33.2% latency for only 0.2 points of gain.
>
> We hope our response has helped clarify these points and we would appreciate it if you could consider raising your score.

---

> > ### Author Rebuttal · Reviewer_9C6n · 2026-04-05
> >
> > I disagree that the answer to Q1 addresses my question: you can always trade-off accuracy against compute/latency by involving less positions (to change) at each step. Hence, it is in principle easy to address the accuracy loss due to factorization by only sampling one token at a time, at the extreme. This is not a limitation of neural nets as such.
> >
> > Thanks for the table, but it does not show the case of 512 steps (which if I understand well means that only one token is generated at each step). The table is encouraging, so I am raising my scores.

---

> > > ### Author Response · Authors · 2026-04-06
> > >
> > > We thank the reviewer for the follow-up. We will clarify that our work specifically targets the parallel generation regime.
> > >
> > > The primary reason we evaluated the models with 64-256 diffusion steps is that both the LLaDA and Dream baselines use 256 or fewer steps by default. Under these settings, CoDD matches or exceeds the performance of the computationally intensive RL baseline while requiring a fraction of the training cost. Regarding the 512-step case, we agree that this setting essentially approaches sequential generation, where the factorization barrier is naturally less of a bottleneck.

---

### Official Review · Reviewer_sN9k · 2026-03-12

**Soundness:** 3
**Presentation:** 3
**Significance:** 3
**Originality:** 3
**Overall Recommendation:** 4
**Confidence:** 4

**Summary:**

This paper investigates the “factorization barrier” present in discrete diffusion language models (dLLMs). In standard dLLMs, tokens predicted at the same time step are treated as independent of one another, creating a trade-off between the speed of parallel generation and the preservation of semantic coherence. The authors contend that this issue arises from a structural misspecification in the modeling assumption rather than from limitations of the underlying model backbone. To address this problem, they introduce Coupled Discrete Diffusion (CoDD), a framework that replaces the fully factorized output distribution with a Probabilistic Circuit (PC) inference layer. By integrating this component, the method can model complex joint dependencies while still maintaining computational efficiency.

**Compliance With Llm Reviewing Policy:**

Affirmed.

**Final Justification:**

My concerns have been addressed.

**Key Questions For Authors:**

- The paper describes the use of **Dynamic Windowing** during the inference stage to cover target tokens with local PC windows. However, the training procedure is described as operating on a "fixed dataset of question-solution pairs" where noise levels are sampled to yield partially observed sequences. It remains unclear whether the PC was trained using the same dynamic windowing cover strategy employed during generation, or if it was trained on isolated, fixed-size segments. Clarification on how the training-time data processing aligns with the inference-time windowing would ensure that the performance gains are not being undermined by a distribution shift.
- How does CoDD handle dependencies that span across two different PC windows? If the windows don't overlap, doesn’t it lose the very "global refinement" that diffusion models are supposed to provide?

**Strengths And Weaknesses:**

**Strengths**

- The paper uses Probabilistic Circuits to resolve the integration bottleneck is a principled and novel application.
- CoDD is highly efficient, requiring only ~3 GPU hours to train (less than 2% of RL-based methods). It introduces minimal latency overhead during inference.
- The framework consistently improves performance across various architectures (LLaDA, Dream) and benchmarks.

---
**Weaknesses**

- The current implementation uses a static PC, which may not fully capture the evolving dependency structures of language across different noise levels. Additionally, th architecture choice of the PC seems not fully discussed.
- Tractable PCs are often limited to shorter contexts compared to global Transformer backbones, requiring the use of Dynamic Windowing to handle full sequences.
- The structural prior $p_{\omega}(x)$ is instantiated as a Probabilistic Circuit with a Hidden Markov Model (HMM) structure and a hidden state size of $N=1024$. While HMMs are computationally efficient, they are traditionally limited to sequential, local dependencies. Given that the primary motivation for CoDD is to capture "complex joint dependencies" that the factorized backbone misses, there is a question regarding whether an HMM structure possesses sufficient expressivity to model the non-local, multi-modal relationships inherent in natural language reasoning tasks
- The proposed framework employs an Adaptive Activation strategy where the Probabilistic Circuit (PC) is only activated when the mask ratio falls below a specific threshold $\gamma$. While Figure 2 illustrates a performance crossover in conditional log-likelihood at a mask ratio of approximately $0.8$, the paper lacks a detailed ablation study or sensitivity analysis on how varying $\gamma$ affects final downstream task performance across different benchmarks.

---

> ### Author Rebuttal · Authors · 2026-03-31
>
> We thank the reviewer for recognizing CoDD's principled use of PCs, its training efficiency, and its consistent improvements across architectures and benchmarks.
>
> > W1:  Static PC may not fully capture the evolving dependency structures of language across different noise levels. Discussion of architecture choice of the PC.
>
> While PC parameters are fixed after training, the inferred dependencies are highly dynamic and explicitly adapt to different noise levels through conditioning.
>
> Specifically, in our formulation $p(x|x_t) \propto p_{\omega}(x|x_t) p_{\theta}(x)$, both the PC and backbone condition on $x_t$, which encodes all observed tokens regardless of the noise level. Because the PC supports exact and efficient marginalization, it dynamically refines its predictions by conditioning on any subset of observed tokens via exact marginalization (Section 4.2). Therefore, rather than relying on a static dependency structure, the PC implicitly and continuously adapts to the evolving generation process.
>
> Empirically, the PC achieves improved conditional log-likelihood across most mask ratios (Figures 1 and 2), demonstrating its ability to generalize across different noise levels. We note that this can be further improved by advancements in PC modeling and learning.
>
> On architecture: the PC is instantiated as an HMM with hidden state size N = 1024 (Section 6.2), chosen for tractable inference and efficiency. Formal definitions and algorithms are in Appendix C and D. We discuss expressivity in W3 below. Notably, our framework is not architecturally constrained to HMMs, other PC instances (tensor networks, hierarchical mixtures) are directly supported and left for future work.
>
> > W2/W3: PCs are limited to shorter contexts and HMMs to local dependencies; is this expressive enough?
>
> Yes, this localized scope is fully expressive for our specific use case, and the context length of 32 is a design choice aligned with the backbone's generation process. Specifically, because LLaDA uses block diffusion with a block size of 32, the model does not simultaneously sample tokens that are more than 32 positions apart. The PC's role is strictly to resolve dependencies between these simultaneously generated tokens, while the LLM backbone handles the global context. In the case of Dream, when predicting 2 or 4 tokens simultaneously, over 90% of the co-sampled tokens naturally fall within a 32-token distance. Therefore, a PC over 32 variables is both highly efficient and perfectly expressive enough to capture the necessary local joint distributions. Note that we can train PCs on longer sequences if needed.
>
> > W4: the paper lacks a detailed ablation study or sensitivity analysis on how varying affects final downstream task performance across different benchmarks.
>
> Ablation studies on both the adaptive activation threshold $\gamma$ and the temperature $\tau$ are provided in Appendix B. We correct Table 4's label as GPQA instead of MATH500. We apologize for the typo.
>
> > Q1: Was the PC trained with the same windowing strategy used at inference? Could there be a distribution shift?
>
> There is no distribution shift as training and inference are consistent with respect to windowing as the PC is trained on 32-token blocks (Section 5.2). For each training sequence, we sample a noise level and mask a portion of the solution tokens; the PC then maximizes the CLL of the masked tokens within 32-token windows. At inference time, Dynamic Windowing places windows of the same size over masked tokens. The "dynamic" aspect refers to the placement of windows (adaptively chosen to cover a subset of masked tokens at each step), not to any change in window size or PC architecture. Since the window size is consistent between training and inference, the PC is applied in exactly the distribution it was trained on.
>
> > Q2: How does CoDD handle dependencies across PC windows? Don't non-overlapping windows lose "global refinement"?
>
> As noted earlier, LLaDA and Dream rarely sample tokens in a single step that are more than 32 tokens apart. Therefore, explicitly modeling joint dependencies across different PC windows during a single step is largely unnecessary. The "global refinement" in diffusion comes from the iterative multi-step process guided by the backbone. The PC's role is to correct the local independence assumptions among simultaneously predicted tokens within a window.
>
> We hope our response has helped clarify these points and we would appreciate it if you could consider raising your score.

---

> > ### Author Rebuttal · Reviewer_sN9k · 2026-04-03
> >
> > Thanks for the response. I will maintain my score.

---

### Official Review · Reviewer_QT9b · 2026-03-12

**Soundness:** 2
**Presentation:** 3
**Significance:** 2
**Originality:** 3
**Overall Recommendation:** 3
**Confidence:** 4

**Summary:**

This work proposes CoDD, a framework for breaking the assumption in dLLM generation that tokens generated in parallel are independent. CoDD introduces a lightweight probabilistic circuit layer to dLLM’s model architecture, which captures complex dependencies between tokens that can be generated in parallel. Evaluation demonstrates that CoDD significantly improves the model accuracy of dLLMs like LLaDA and Dream. CoDD also presents generation collapse when using fewer generation steps and greater parallel generation degree. The training cost of CoDD is also small, making it easy to be deployed.

**Compliance With Llm Reviewing Policy:**

Affirmed.

**Key Questions For Authors:**

1. How can we adapt the probabilistic circuit module with different noise levels and flexible dependencies? Is it feasible by only modifying the module? Or do we need to propose different formulations of this module?
2. How do we position CoDD with models with , e.g., d1, d2. How do we compare with speedup approaches, like SeeDiffusion?
3. How does CoDD perform on more general generation tasks, compared to its strong improvement on structured math and coding tasks?

**Limitations:**

yes

**Strengths And Weaknesses:**

**Strengths**

1. The proposed CoDD is flexible and collaborates well across different models (LLaDA and Dream) and different generation paradigms (block-wise diffusion and conventional bidirectional diffusion). In addition, CoDD acts as a plugin added to frozen baseline models, demonstrating lower training cost than fine-tuning or RL.
2. CoDD exhibits robust generation at a few steps and avoids accuracy drop, enabling faster inference without sacrificing the model performance.
3. The formulation of joint token probabilities and the theoretical foundation of the probabilistic circuit module is sound.

**Weakness**

1. The dependency that the probabilistic circuit module learns is static, and is an average dependency, which cannot adapt to different noise levels and generation progress.
2. The any-order autoregressive sampling method introduces a noticeable latency overhead to the generation process.
3. The evaluation heavily focuses on structured reasoning and coding tasks (MATH, GSM8K, MBPP), making its effectiveness on general open-ended text generation unclear.

---

> ### Author Rebuttal · Authors · 2026-03-31
>
> We thank the reviewer for recognizing CoDD's flexibility across models and generation paradigms, its robust few-step generation, and its sound theoretical foundation.
>
> > W1/Q1: The PC learns a static, average dependency that cannot adapt to different noise levels.
>
> While the parameters of a PC are fixed after training (as with any neural network), the learned dependencies do adapt to different noise levels through conditioning. The unconditional product composition $​​p(x) \propto p_{\omega}(x) \cdot p_{\theta}(x)$ becomes $p(x|x_t) \propto p_{\omega}(x|x_t) \cdot p_{\theta}(x)$ when context is provided, where both the PC and backbone condition on $x_t$, which encodes the current noise level and which tokens are observed. The PC can further condition on any subset of currently observed tokens via exact marginalization (Section 4.2), refining predictions as tokens are revealed. So the PC implicitly adapts to different noise levels and generation progress through conditioning, despite having fixed parameters.
>
> Our Adaptive Activation strategy (Section 5.3) further specializes in the PC's guidance. As shown in Figure 2, the PC provides the most benefit at low mask ratios, where local token correlations are the dominant source of incoherence. It achieves improved conditional log-likelihood across all mask ratios below 0.8, demonstrating that it generalizes across the diffusion trajectory. We activate the PC only when the mask ratio falls below this threshold, targeting precisely the noise levels where it helps the most.
>
> For even more flexible dependency structures, Appendix E outlines dynamic structural selection, where control parameters from the backbone could reweight sub-distributions within the PC for different noise regimes, which we believe is a promising future direction enabled by CoDD's modular design.
>
> > W2: The any-order autoregressive sampling method introduces a noticeable latency overhead to the generation process.
>
> The Any-Order (AO) sampling overhead is noticeable relative to base CoDD, but remains significantly lower than the RL-based alternatives while also achieving higher accuracy. As shown in Table 3, CoDD+AO introduces a 26.92% latency overhead on MATH500 at 64 steps, whereas diffu-GRPO incurs 33.22% overhead.
>
> Further, CoDD+AO also achieves higher accuracy (17.0% vs. 15.8% at 64 steps, Figure 3) and costs orders of magnitude less to train (~3 GPU hours vs. >150 GPU hours for d-GRPO). AO is best understood as an optional precision-enhancing mode when modest additional latency is acceptable in exchange for further quality improvement.
>
> > W3/Q3: Performance on general open-ended text generation
>
> To evaluate the generalization of CoDD beyond reasoning tasks, we assessed its performance on open-ended text generation using the WikiText validation set. For each evaluation instance, we randomly truncated a sequence from the validation set to serve as a conditioning prefix. The models were then tasked with generating the continuation of the text. We trained the PC using a corpus of samples generated by a pre-trained GPT-2 model. The base model is Dream with low-confidence decoding.
>
> | Steps | Method       |  PPL  |  Div  | Rep-2 | Rep-4 | NLL/tok |
> |-------|-------------|-------|-------|-------|-------|---------|
> | 16    | Dream    | 3.32  | 0.217 | 0.783 | 0.707 | 1.2013  |
> | 16    | + CoDD      | 2.92  | 0.188 | 0.812 | 0.747 | 1.0707  |
> | 32    | Dream    | 4.67  | 0.341 | 0.659 | 0.571 | 1.5403  |
> | 32    | + CoDD      | 4.08  | 0.316 | 0.684 | 0.599 | 1.4060  |
> | 64    | Dream    | 4.66  | 0.484 | 0.516 | 0.401 | 1.5397  |
> | 64    | + CoDD      | 4.15  | 0.463 | 0.537 | 0.427 | 1.4228  |
>
> We demonstrate that it consistently improves sequence likelihood and perplexity across all step budgets, while incurring a minimal ~5% latency overhead.
>
> > Q2: How do we position CoDD with models with , e.g., d1, d2. How do we compare with speedup approaches, like SeeDiffusion?
>
> CoDD is orthogonal to d1 and d2, which improved dLLM reasoning through RL-based fine-tuning of the neural backbone. CoDD instead operates at the output distribution level and requires no backbone changes, addressing a complementary problem: RL methods teach the model what to reason, while CoDD corrects the structural misspecification in how the model decodes. In principle, CoDD can be applied on top of d1/d2, which we consider an exciting direction for future work.
>
> We were unable to find "SeeDiffusion" by this exact name. We would appreciate it if the reviewer could point us to the specific paper so we can provide a more precise comparison.
>
> We hope our response has helped clarify these points and we would appreciate it if you could consider raising your score.

---

> > ### Author Rebuttal · Reviewer_QT9b · 2026-04-04
> >
> > The SeeDiffusion should be Seed Diffusion (Seed Diffusion: A Large-Scale Diffusion Language Model with High-Speed Inference).
> > In general your response is good, just to clarify this incorrect paper name by my mistake.

---

> > > ### Author Response · Authors · 2026-04-05
> > >
> > > We appreciate the reviewer's suggestion to position CoDD alongside Seed Diffusion. Both approaches share the goal of enabling efficient, high-quality, parallel generation, but they operate on complementary levels.
> > >
> > > As a plug-and-play post-training framework, CoDD is orthogonal to training techniques like Seed Diffusion and can be directly integrated into their regimes. Since Seed Diffusion also relies on a standard Transformer backbone (which assumes simultaneously predicted tokens are mutually independent), CoDD can augment the backbone with a Probabilistic Circuits (PC) layer.
> > >
> > > With the joint model, we can re-derive the ELBO that Seed Diffusion uses without the independence assumption by substituting the denoising distribution $p_{\theta}(x_0|x_t)$ with the joint distribution $p_{\theta,\omega}(x_0|x_t) \propto p_{\omega}(x_0) \cdot p_{\theta}(x_0|x_t)$. By training a valid evidence lower-bound interpretation under this joint model, CoDD allows Seed Diffusion's on-policy learning and constrained-order trajectories to optimize for a distribution that is structurally capable of capturing the inter-token dependencies inherent in language.

---

### Decision · Program_Chairs · 2026-04-30

**Decision:**

Accept (regular)

**Comment:**

The paper addresses the speed-quality trade-off of parallel token prediction using DLMs with the proposed method CoDD. CoDD augments a diffusion LM with a tractable probabilistic inference layer to model richer joint dependencies among parallel predictions.
The method is reported to improve both reasoning accuracy and few-step robustness across multiple DLM backbones, while retaining most of the efficiency benefits of parallel decoding.

The core technical idea of adding a lightweight probabilistic layer on top of frozen diffusion backbones is clean and technically sound. CoDD works across different backbones, including LLaDA and Dream, and across different generation paradigms such as block-wise and bidirectional diffusion. Moreover, the method is efficient as the cost can be as low as around 3 GPU hours in training.

The major concerns of this paper are whether the added structure is expressive enough for the full dynamics of diffusion decoding, the approach may introduce additional latency overhead at inference, and the task scope is limited to MATH, GSM8K, and MBP. These points are addressed during rebuttal.

Overall, the paper is satisfactory and reviewers agree in the acceptance.